# The Contributions to the Explosive Growth of PM$_{2.5}$ Mass due to Aerosols-Radiation Feedback and Further Decrease in Turbulent Diffusion during a Red-alert Heavy Haze in Jing-Jin-Ji in China

Hong Wang[1,2] [*], Yue Peng[1,2], Xiaoye Zhang[1,3*], Hongli Liu[1], Meng Zhang[4], Huizheng Che[1], Yanli Cheng[1], Yu Zheng[1,2]

Author list:
Hong Wang, wangh@cma.gov.cn,
Yue Peng, 1131509950@qq.com; nuist_PY@163.com
Xiaoye Zhang, xiaoye@cma.gov.cn
Hongli Liu, liuhl@cma.gov.cn
Meng Zhang, 316398453@qq.com
Huizheng Che, chehz@cma.gov.cn
Yanli Cheng, chengyl@cma.gov.cn
Yu Zheng, hblfzhengyu@126.com

# The Contributions to the Explosive Growth of $PM_{2.5}$ Mass due to Aerosols-Radiation Feedback and Further Decrease in Turbulent Diffusion during a Red-alert Heavy Haze in Jing-Jin-Ji in China

Hong Wang[1,2] *, Yue Peng[1,2], Xiaoye Zhang[1,3*], Hongli Liu[1], Meng Zhang[4], Huizheng Che[1], Yanli Cheng[1], Yu Zheng[1,2]

State Key Laboratory of Severe Weather (LASW), Chinese Academy of Meteorological Sciences (CAMS), CMA, Beijing 100081, China
Collaborative Innovation Center on Forecast and Evaluation of Meteorological Disasters, Nanjing University of Information Science & Technology, Nanjing 210044, China
Center for Excellence in Regional Atmospheric Environment, Institute of Urban Environment, Chinese Academy of Sciences (CAS), Xiamen 361021, China
Beijing Meteorological Bureau, Beijing 100089, China

Correspondence to: Hong Wang (wangh@cma.gov.cn),Xiaoye Zhang (xiaoye@ cma.gov.cn)

**Abstract.** The explosive growth of $PM_{2.5}$ mass usually results in extreme $PM_{2.5}$ levels and severe haze pollution in East China, and is generally underestimated by current atmospheric chemistry models. Based on one such model, GRPAES_CUACE, three sensitivity experiments – a "background" experiment (EXP1), "online aerosol feedback" experiment (EXP2), and an "80% decrease in turbulent diffusion coefficient" of chemical tracers" experiment, based on EXP2 (EXP3) – were designed to study the contributions of aerosol–radiation feedback (AF) and decrease in turbulent diffusion coefficient to the explosive growth of $PM_{2.5}$ during a "red-alert" heavy haze event in China's Jing–Jin–Ji region. The results showed that the turbulent diffusion coefficient calculated by EXP1 was about 60–70 $m^2$/s on the clear day and 30–35 $m^2$/s on the haze day. This difference in diffusion coefficient was not enough to distinguish between the unstable atmosphere on the clear day and extremely stable atmosphere during the $PM_{2.5}$ explosive growth stage. Also, the inversion calculated by EXP1 was obviously weaker than the actual inversion from sounding observations on the haze day. This led to a 40%–51% underestimation of $PM_{2.5}$ by EXP1; AF reduced by about 43%–57% diffusion coefficient during the $PM_{2.5}$ explosive growth stage, which strengthened the local inversion obviously; plus, the local inversion indicated by EXP2 was much closer to the sounding

observations than that by EXP1. This resulted in a 20%–25% reduction of $PM_{2.5}$ negative errors in the
model, reaching as low as −16% to −11% in EXP2. However, the inversion produced by EXP2 was still
weaker than the actual observation, and AF could not solve all the problems of $PM_{2.5}$ underestimation.
Based on EXP2, the 80% decrease in turbulent diffusion coefficient of chemical tracers in EXP3 resulted in
near-zero turbulent diffusion, referred to as an "turbulent intermittence" atmospheric state, which resulted
in a further 14%–20% reduction in $PM_{2.5}$ underestimation, and the negative $PM_{2.5}$ errors were reduced to
−11% to 2%. The combined effects of AF and decrease in turbulent diffusion coefficient solved over 79%
of the underestimation of the explosive growth of $PM_{2.5}$ in this study. The results show that online
calculation of AF is essential for the prediction of $PM_{2.5}$ explosive growth and peaks during severe haze in
China's Jing–Jin–Ji region. Besides, an improving in the planetary boundary layer scheme with respect to
extremely stable atmospheric stratification is also essential for a reasonable description of local "turbulent
intermittence" and a more accurate prediction of $PM_{2.5}$ explosive growth during severe haze in in this
region of China.
**Keywords:** aerosol–radiation feedback; turbulent diffusion; planetary boundary layer scheme; temperature
inversion; $PM_{2.5}$

**1 Introduction**

Since 2013, East China has been experiencing unprecedented intrusions of severe haze accompanied by high levels of particulate matter (PM) of less than 2.5 microns in aerodynamic diameter ($PM_{2.5}$), causing wide public concern (Ding et al., 2013; Wang et al. 2013; Huang et al., 2014; Wang et al., 2014; Sun et al., 2014; Hua et al., 2016; Yang et al., 2015; Zhong et al., 2017, 2018a, 2018b). The instantaneous $PM_{2.5}$ concentration is usually in the hundreds of ug/m$^3$ during severe haze episodes, occasionally exceeding one thousand, in the metropolitan region of Beijing–Tianjin–Hebei, referred to here as Jing–Jin–Ji, and its surroundings of East Shanxi, West Shandong, and North Henan in East China (Wang et al., 2014; Quan et al., 2014; Sun et al., 2014; Yang et al., 2015; Zheng et al., 2016). Studies have shown, however, that models generally underestimate the explosive growth and peak values of $PM_{2.5}$ during severe hazes, especially in Jing–Jin–Ji (Wang et al., 2013; Wang et al., 2014; Li et al., 2016).

The causes of $PM_{2.5}$ explosive growth and its underestimation by atmospheric chemistry models are complex and uncertain at present, but it possibly involves local emissions, reginal transportation, aerosol physicochemical processes, gas–particle conversion, meteorological conditions, and so on. However, the actual atmospheric stability and how accurate it is described by atmospheric models is a fundamental problem that cannot be ignored among others. Local or regional meteorological conditions dictate whether haze occurs and what the $PM_{2.5}$ level may be (Zhang et al., 2014; Zheng et al., 2015; Gao et al., 2016) when source emissions are unchanged for a short period of time. The meteorological conditions of the planetary boundary layer (PBL) are a key and direct trigger for the emergence of a haze event (Wang et al., 2014; Li et al., 2016; Zhong et al., 2017). Turbulent diffusion is an important factor to characterize PBL meteorology when the atmosphere is stable. Also, it is a major pathway of particle and gaseous pollutant exchange from the surface to upper atmosphere; and when haze occurs, pollutant dispersal via the upper-level winds can take place when haze is accompanied by calm surface winds and weak vertical motion of air in surface layers and the PBL. The intensity of turbulent diffusion largely determines the severity of haze pollution. Thus, a reasonable description of turbulent diffusion by PBL schemes in atmospheric chemistry models is vital for the prediction of severe pollution (Hong et al., 2006; Wang et al., 2015; Hu et al., 2012, 2013a, 2013b; Li et al., 2016). The latest studies in this field of research show (Wang et al., 2015; Li et al., 2016) that current PBL schemes may be insufficient for describing the extremely weak turbulent diffusion

conditions when extremely severe haze occurs in Jing–Jin–Ji, which more broadly may be one important
reason why $PM_{2.5}$ peaks are underestimated by atmospheric chemistry models. More specifically, there may
be two independent reasons why the description of extremely weak turbulent diffusion in atmospheric
models is deficient. One is that aerosol–radiation feedback (AF) is not calculated online in the model run.
AF may restrain turbulence by cooling the surface and PBL while heating the atmosphere above it when
aerosols with certain absorption characteristics are concentrated in the PBL (Wang et al., 2010; Forkel et al.,
2012; Gao et al., 2014, 2015; Wang et al., 2015; Ding et al., 2016; Li et al., 2016; Miao et al., 2016; Petaja
et al., 2016; Gao et al., 2017; Qiu et al., 2017; Zhong et al., 2018). Ignoring AF is likely to lead to an
obvious overestimation of turbulent diffusion when the $PM_{2.5}$ concentration exceeds a certain value, which
is worthy of further study. The other possible reason is that the extremely weak turbulence resulting in
extremely severe haze is not fully described by the atmospheric chemistry model (Li et al., 2016).
In the present work, a "red-alert" heavy haze event (issued by China's Ministry of Environmental
Protection when the air pollution index is forecast to exceed 300 over the next three days) that occurred
during 15–23 December 2016 in China's Jing–Jin–Ji region was selected to study the contributing factors to
$PM_{2.5}$ explosive growth and peaks, and the possible deficiency of atmospheric models in describing
extremely weak turbulent diffusion.
**2 Model, data and methods**
**2.1 Model**
Focusing on dust and haze pollution in China and East Asia, the Chinese Unified Atmospheric
Chemistry Environment (CUACE) (Gong and Zhang, 2008) was online-integrated into the mesoscale
version of the Global/Regional Assimilation and PrEdiction System (GRAPES_meso), developed by the
Chinese Academy of Meteorological Sciences (Chen et. al., 2008; Zhang and Shen, 2008), to build an
online chemical weather forecasting model, GRAPES_CUACE (Wang et al., 2009, 2010; 2015a; Zhou et
al., 2012). The main components of GRAPES_CAUCE include: a model dynamic core; a modularized
physics package (Xu et al., 2008); an atmospheric chemistry module, CUCAE, with online coupling of
direct and indirect aerosol feedback; and an emissions inventory. The dynamic framework of
GRAPES_CUACE is semi-implicit, semi-Lagrangian, fully compressible, and non-hydrostatic (Yang et al.,
2007, 2008; Chen et al., 2008). A height-based terrain-following coordinate system is used, and there are 33
vertical layers form the surface to 30 km. A longitude–latitude grid is adopted in the spatial discretization of
the model and the horizontal resolution may vary upon request. The physics package can also be tailored by
the user (Xu et al., 2008), and Table 1 lists the specific physics and chemistry schemes used in this study.
The gas-phase chemistry of RAD II (Stockwell et al., 1990), with 63 gaseous species through 21
photochemical reactions and 121 gas-phase reactions, is used in this study. The aerosols include sea salt
(SS), sand/dust (SD), black carbon (BC), organic carbon (OC), sulfates (SFs), nitrates (NI) and ammonium
salts (AM), and aerosol processes involving hygroscopic growth, coagulation, nucleation, condensation,
dry and wet deposition, scavenging, aerosol activation, and so on. The formation of SF aerosols and
secondary organic aerosols from gases, NI and ammonium formed through gaseous oxidation, and
ISORROPIA (Fountoukis et al., 2007) calculating the thermodynamic equilibrium between NI and
ammonium and their gas precursors, are considered in CAUCE, which has been evaluated and introduced
in previous studies (Gong and Zhang et al., 2008; Zhou et al., 2008, 2012).

Based on the modeled aerosol concentrations, vertical profiles of temperature change, including direct

aerosol impacts, are calculated by the radiation model and fed back online to the model dynamic core at
each grid point and every time step, which reforms the model temperature field, dynamic process, regional
circulation and meteorological conditions, in turn ultimately impacting the aerosol concentration. The
external mixing of aerosols species (SS, SD, BC, OC, SF, NI, AM) and particle size bins are used in the
calculation of AF, as introduced and evaluated in detail in previous studies (Wang et al., 2009, 2010, 2015a,
2015b). With this two-way GRAPES_CUACE model, aerosol–radiation–PBL–meteorological interactions,
as well as aerosol–cloud–precipitation interactions and regional pollution and transportation of $PM_{2.5}$ etc.,
have been successfully studied (Wang et al., 2010, 2015a, 2015b; Zhou et al., 2012, 2016; Jiang et al., 2015;
Zhang et al., 2018).

The turbulent diffusion coefficient is calculated by the YonSei University PBL scheme (Hong et al.,

2006), which is a revised vertical diffusion package based on the nonlocal boundary layer vertical diffusion
scheme in a medium-range forecast (MRF) model (Hong et al., 1996). The major ingredient of the revision
is the inclusion of an explicit treatment of entrainment processes at the top of the PBL, compared with the
MRF PBL scheme. The specific calculation method of diffusion coefficient is shown in Hong et al. (1996),
and has been selected as a standard option in MRF models (Caplan et al. 1997; Farfán and Zehnder, 2001;
Basu, et al., 2002; Bright and Mullen, 2002; Mass et al., 2002) as well as the Weather Research and
Forecasting model (Hong et al., 2006) in the National Centers for Environmental Prediction (NCEP) since
its establishment.
The horizontal resolution of the model adopted here was $0.15° \times 0.15°$, to match the resolution of the
emission source. Considering the impacts of the interregional transport of pollutants, East China (100°–
140°E, 20°–60°N) (Figure 1a) was set as the model domain, but our discussion focuses mainly on the most
polluted area, Jing–Jin–Ji (red frame in Figure 1a), for which Figure 1b illustrates the geographical and
topographical features. There are two balloon sounding stations, Xingtai and Beijing (yellow stars in Figure
1b) in our study area. Xingtai, located in southern Hebei province at the eastern foot of the Taihang
Mountains, is influenced by descending airflow from the mountains in winter, and in recent years has
frequently been ranked the most polluted city in China. The topography of Xingtai and the serious haze
pollution it experiences are closely related to its situation on the southern plain of Jing–Jin–Ji. Beijing,
located next to Tianjin and surrounded by Hebei, lies in the transitional zone from the Yan Mountains to its
southern plain, and represents the most polluted areas in the central part of Jing–Jin–Ji.
**2.2  Emissions inventory**
Based on the Multi-resolution Emissions Inventory for China in 2012 (He et al., 2012), the changes in
East China of five kinds of emission sources – industrial, domestic, agricultural, natural, and traffic – were
obtained from national statistical data with respect to industry, energy consumption, road networks, and
motor vehicles, and updated to 2015 and 2016. Five reactive gases ($SO_2$, NO, $NO_2$, CO, $NH_3$), 20 volatile
organic compounds [VOCs (ALD, $CH_4$, CSL, ETH, $HC_3$, $HC_5$, $HC_8$, HCHO, ISOP, KET, NR, $OL_2$, OLE,
OLI, OLT, $ORA_2$, PAR, TERPB, TOL, XYL), listed in Table 2], and five aerosol species (BC, OC, SF, NI
and fugitive dust), were obtained via the above emissions data according to the input requirement of the
CUACE model. The horizontal grid resolution was $0.15° \times 0.15°$ and there was one emissions dataset for
each month at hourly intervals.
**2.3 Data**
Hourly observational $PM_{2.5}$ concentration data for more than 1440 surface observational stations (blue
dots in Figure 1) from the China National Environmental Monitoring Centre (http://www.cnemc.cn) during
15–23 December 2016 were used to evaluate the model results. The hourly observational meteorological
data, including wind speed and temperature, from 500 surface automatic observation stations of the China
Meteorological Administration (CMA) in the Jing–Jin–Ji region (red triangle in Figure 1b), were used for
model validation. Meteorological balloon sounding data from the CMA at 0000 UTC (early morning) and
1200 UTC (dusk, local time) in Beijing and Xingtai (yellow star in Figure 1b) during the same period were
also used to compare with the modeled results. There is one AERONET station (Holben et al., 1998),
Xianghe, and two CARSNET stations (Che et al., 2009; 2014; 2015), Beijing and Shijiazhuang, in the
Jing–Jin–Ji region (black crosses in Figure 1b). Observed aerosol optical depth (AOD) and single scattering
albedo (SSA) data from these three stations during the same period were also used for model evaluation.
NCEP $0.25° \times 0.25°$ global analysis gridded data (https://rda.ucar.edu/datasets/ds083.3) were used as the
model's initial and six-hourly lateral boundary meteorological input fields. The initial values of chemical
tracers were obtained according to their five-year mean climatic values. The results of the first 120 hours of
the model were discarded to eliminate the effects of the chemical initial fields.
**2.4 Experimental design**
Both dynamic processes of the regional atmosphere and solar radiation have important impacts on
turbulent diffusion and PBL processes. When severe haze occurs, it has been showed from observation
study (Zhong et al., 2018) that surface-level daily direct radiative exposure is reduced by around 89%
compared with clean days, suggesting the possibility of a huge difference in turbulent diffusion between
severe haze and clean days. However, it is difficult to distinguish between the two reasons for extremely
weak turbulent diffusion in the true atmosphere, because of the complicated relationship between
atmospheric dynamics and solar radiation. However, meaningful results might be possible by conducting
sensitivity experiments using an atmospheric chemistry model. Here, three such experiments (EXP1, EXP2,
and EXP3 – see Table 3 for descriptions) were designed to discuss the contributing factors to extremely
weak turbulence and corresponding $PM_{2.5}$ explosive growth, along with the insufficient description of
extremely weak turbulent diffusion by PBL schemes in atmospheric chemistry models. All other model
dynamic processes, physical options, and initial input data of the meteorology and chemical tracers were
same for the three experiments, i.e., except the differences shown in Table 3. In EXP3, a further decrease in
the turbulent diffusion coefficient based on EXP2 was only applied to the diffusion coefficient of chemical
tracers in CUACE mode; the diffusion coefficient in other physical packages and the dynamic framework
of GRAPES_MESO was the same as in EXP1 and EXP2.
**3 Results and discussion**
The studied haze episode began on 15 December 2016. $PM_{2.5}$ began to gather and climb slowly at this
time, but was below 150 ug/m$^3$ in most of Jing–Jin–Ji from 00:00 UTC 15 to 00:00 UTC 17 December – a
period we refer to as the "climbing stage" of $PM_{2.5}$. From 00:00 UTC 17 to 00:00 UTC 21 December, $PM_{2.5}$
increased rapidly, and reaching a peak of 400–600 ug/m$^3$ in most of the study area. We refer to this period
as the "explosive growth stage" of $PM_{2.5}$. In this section, we focus mainly on the contributions of AF and
decrease in turbulent diffusion coefficient to the $PM_{2.5}$ during this stage.
**3.1 Synoptic background**
The circulation in the upper atmosphere and the surface-level synoptic system controlling Jing–Jin–Ji
remained relatively stable during the maintenance of this haze episode. Figure 2 displays the geopotential
height, temperature, and winds in the upper (500 hPa), middle (700 hPa) and lower (850 hPa) atmosphere,
as well as PBL levels (900, 950, 1000 hPa), at 0000 UTC 19 December 2016, to show the meteorological
background. It can be seen that the geopotential height in the upper atmosphere (500 hPa) showed zonal
circulation in East Asia. There was a horizontal trough north of Jing–Jin–Ji (black frame) in the upper and
middle atmosphere (500 and 700 hPa), and the region was controlled by moderate northwesterly or
westerly air flow at the bottom of the trough. The temperature and wind fields at 500 and 700 hPa both
showed that cold air in the upper and middle atmosphere was weak. The 850-hPa geopotential height
showed that the subtropical high in the East Sea was strong; also, Jing–Jin–Ji was in the pressure
equalization field to the northwest periphery of the subtropical high and the wind was very weak at this
level due to the blocking of the subtropical high. The 900-, 950- and 100-hPa geopotential heights all
showed that Jing–Jin–Ji was located in the pressure equalization field between the "northwest land high"
and southeast subtropical high within the whole PBL, and the land high was weaker than the subtropical
high. This resulted in a small pressure gradient, weak and thin wind fields, and a stable atmospheric
situation within the PBL, which was conducive to the maintenance of the haze episode.
**3.2 Observation–model comparison**
Meteorological factors not only at the surface but also in the PBL are key in affecting haze processes
and $PM_{2.5}$ concentrations (Wang et al., 2014a, 2014b). Unfortunately, however, most numerical models
struggle to simulate these aspects, which is also a key point determining the performance of atmospheric
chemistry models (Hu et al., 2013a, 2013b; Li et al., 2016).
Using hourly meteorological data from surface automatic observation stations of the CMA, the
surface wind speed and temperature at Beijing and Xingtai, and the average for Jing–Jin–Ji, according to
the results of EXP1, EXP2 and EXP3, were evaluated for the period 15–24 December 2016 (Figure 3). It
can be seen that, in Beijing, the modeled surface wind speed in the three experiments was in good
agreement with observation, in terms of the overall trend as well as the maximum and minimum values.
The observed and modeled wind speed was basically below 2 m/s during 17–21 December (i.e., the
explosive growth stage of $PM_{2.5}$). The modeled wind speed at Xingtai was slightly worse than that at
Beijing, but the overall trend of change was basically consistent with observation, and the wind speed was
also below 2 m/s during the explosive growth stage. The modeled wind speed was to an extent higher than
observed at the beginning and end in Xingtai. The trend of change in the modeled average wind speed for
the Jing–Jin–Ji region showed reasonable agreement with observation and was closest to the observed
situation in the explosive growth stage. In general, the modeled regional wind was higher than observed.
Comparison of the wind speed among the three experiments showed that the wind speeds in EXP2 and
EXP3 were basically same, but to a varying degree both were smaller than in EXP1 at Beijing and Xingtai,
as well as for Jing–Jin–Ji as a whole, during the explosive growth stage, showing that AF decreased the
surface wind speed. The trend of temperature change according to the three experiments was also consistent
with observation, at Beijing, Xingtai, and Jing–Jin–Ji as a whole. However, it was found that the modeled
temperature was obviously higher than observed, especially during the explosive growth stage. The
temperature in EXP2 and EXP3 was basically same, but lower than in EXP1, which was much closer to
observation, indicating that AF reduced the overestimation of surface temperature in Beijing, Xingtai, and
Jing–Jin–Ji as a whole. However, the temperature in EXP2 and EXP3 was also higher than observed during
the explosive growth stage, suggesting a role played by other uncertainties in the PBL scheme besides AF,
which is deserving of more detailed study in the future. Also shown in Figure 3 are the PBL-mean winds of
the three experiments for Beijing, Xingtai, and Jing–Jin–Ji as a whole. Unfortunately, no observational data
were available to evaluate them. However, comparison of the PBL's wind and temperature according to the
three experiments showed that the PBL-mean wind was basically below 4 m/s while the temperature was
high in the explosive stage at Beijing, Xingtai, and in Jing–Jin–Ji as a whole. Similar to the surface-level
results, the PBL-mean wind speed and temperature in EXP2 and EXP3 were basically the same, but the
wind speed in these two experiments was obviously lower than that in EXP1. This indicated that the
reduction in wind speed by AF was more obvious in the PBL than at ground level. Meanwhile, comparison
of the surface-level and PBL temperature of the three experiments showed that the cooling effect of AF was
much stronger at the surface than in the PBL.
Aerosol optical properties, including AOD, SSA and asymmetry factor, largely determine the direct
radiative effects of aerosols. The observed AOD (Table 4) and SSA (Table 5) at Shijiazhuang, Beijing and
Xianghe were used to evaluate the modeled results for the period 15–22 December. Because the differences
in the modeled AOD and SSA results of EXP1, EXP2 and EXP3 were small, those of EXP1 only are
referred to here. The values of modeled AOD and SSA and their temporal trends of change during 15–22
December were basically consistent with observation at Beijing, Shijiazhuang and Xianghe, thus
demonstrating good model performance in terms of its description of aerosol optical properties. Both the
observed and modeled SSA at Shijiazhuang, Beijing and Xianghe (Table 5) showed that the SSA was
obviously higher during the explosive growth stage compared with that at the beginning or end of the haze
on 15–16 and 22 December, illustrating that the scattering characteristics of composite aerosols increase
obviously when high AOD and $PM_{2.5}$ occur on severe haze days in the Jing–Jin–Ji region. The accurate
description of AOD and SSA, especially with respect to the change in SSA from clean to haze days, is the
basis of the following discussion on the effects of aerosols on $PM_{2.5}$.
Figure 4 displays the averaged observed $PM_{2.5}$ ($PM_{2.5}$_OBS) and simulated $PM_{2.5}$ of EXP1
($PM_{2.5}$_EXP1), EXP2 ($PM_{2.5}$_EXP2) and EXP3 ($PM_{2.5}$_EXP3) during the explosive growth stage. It can be
seen from $PM_{2.5}$_OBS results that the averaged $PM_{2.5}$ values generally exceeded 100 μg/m$^3$ in east China,
and Jing–Jin–Ji comprised the most polluted areas with $PM_{2.5}$ reaching 300–400 μg/m$^3$ in parts of Beijing,
Tianjin, central-south Hebei, western Shandong, and northern Henan. The most polluted area with $PM_{2.5}$
values of 500–700 μg/m$^3$ appeared in southern Hebei and northern Henan provinces and the maximum
value of $PM_{2.5}$ even exceeded 700 μg/m$^3$ in part area in southern Hebei. Comparison of $PM_{2.5}$_EXP1 and
$PM_{2.5}$_OBS shows that $PM_{2.5}$_EXP1 was obviously lower than $PM_{2.5}$_OBS on the whole. Notably, EXP1
failed to simulate the $PM_{2.5}$ > 300 μg/m$^3$. $PM_{2.5}$_OBS was approximately 200–300 μg/m$^3$ over most of
Shandong, while $PM_{2.5}$_bk was only 100–200 μg/m$^3$ in this region. Compared with $PM_{2.5}$_EXP1, the
$PM_{2.5}$_EXP2 values were significantly improved by AF, and were much closer to $PM_{2.5}$_OBS. The high
$PM_{2.5}$_OBS centers of 300–400, 400–500 and 500–600 μg/m$^3$ were almost simulated by EXP2, indicating
the important effects of AF in simulating such high values of $PM_{2.5}$. However, the simulated areas of these
centers were smaller than those of $PM_{2.5}$_OBS. EXP2 also failed to simulate the maximum $PM_{2.5}$ values
over 600 μg/m$^3$ observed in southern Hebei. $PM_{2.5}$_EXP3 just about made up for this shortcoming;
compared with $PM_{2.5}$_EXP1 and $PM_{2.5}$_EXP2, $PM_{2.5}$_EXP3 was undoubtedly the closest to $PM_{2.5}$_OBS
both in terms of $PM_{2.5}$ extremes and the area of influence. These findings illustrate that both AF and
decrease in turbulent diffusion coefficient in atmospheric chemistry models are required for the effective
prediction of $PM_{2.5}$ explosive growth during severe haze in China's Jing–Jin–Ji region.
**3.3 Change in downward solar radiation flux by aerosols and decrease in turbulent diffusion**
**coefficient**
PM in the atmosphere will inevitably lead to changes in surface and atmospheric solar radiation flux.
When severe haze occurs, most PM is concentrated in the atmosphere near the surface and within the PBL;
solar radiative flux reaching the ground is reduced greatly, which is a direct trigger for the subsequent
changes in thermodynamics, dynamics, and then atmospheric stratification. Any factor leading to a change
in the atmospheric PM loading might result in a change in the surface downward solar radiation flux
(SDSRF). We calculated the percentage changes in SDSRF (W/m$^2$) between EXP2 and EXP1
[(SDSRF$_{EXP2}$ − SDSRF$_{EXP1}$) / SDSRF$_{EXP1}$], and EXP3 and EXP1 [(SDSRF$_{EXP2}$ − SDSRF$_{EXP1}$) /
SDSRF$_{EXP1}$)], to study the impacts on SDSRF of aerosols and decrease in turbulent diffusion coefficient.
Figure 5 shows the mean percentage change in SDSRF (W/m$^2$) owing to aerosols (a) and aerosols plus
decrease in turbulent diffusion coefficient, during the explosive growth stage. It can be seen that SDSRF
was reduced by more than 50% by aerosols over most of the study region (60%–65% in Jing, Jin, most of Ji,
and northern Shandong, and even 65%–70% in Jing, Jin, and part of Ji), indicating the important influence
of aerosols on SDSRF. Comparison of Figures 5b and 5a shows that this reduction in SDSRF owing to
aerosols (Figure 5a) in EXP2 was further strengthened by the decrease in turbulent diffusion coefficient of
chemical tracers in EXP3 (Figure 5b) in certain regions, because decrease in turbulent diffusion coefficient
led to the accumulation of more PM$_{2.5}$ near the surface (Figure 3), less transport and, subsequently, an
increase in total PM$_{2.5}$ loading. It can also be seen that the difference between Figures 5a and 5b is
negligible. This is because the major impact of decrease in turbulent diffusion coefficient was to reform the
vertical distribution of the atmospheric loading of PM$_{2.5}$, and its impact on the total-column PM$_{2.5}$ was
minor. On the other hand, the reduction in SDSRF owing to aerosol radiation was already considerable, and
so the change in SDSRF owing to the increased total-column PM$_{2.5}$ by decrease in turbulent diffusion
coefficient would be secondary. This value of SDSRF reduction owing to aerosols and decrease in turbulent
diffusion coefficient is basically consistent with the 56%–89% difference of observational radiative
exposure between clear and haze days during the same period (Zhong et al., 2018).
**3.4 Influence of aerosols on the reforming of the local atmospheric temperature profile**
Offline and online studies indicate a reforming of the atmospheric temperature profile owing to the
direct effect of aerosol radiation (Wang et al., 2010, 2015b; Forkel et al., 2012; Gao et al., 2014, 2015;
Wang et al., 2014; Gao et al., 2017; Ding et al., 2016). In our previous work (Wang et al., 2015a, 2015b),
composite aerosol mixing of BC, OC, SF, NI, dust, ammonium, and sea salt aerosols was online coupled
into the GRAPES_CAUCE model. On this basis, in the present study, the changes in the mean temperature
profile of Jing–Jin–Ji during daytime owing to aerosol radiation were calculated for 15–20 December 2016.
It can be seen from Figure 6 that aerosols cooled the atmosphere below 750–800 hPa, but warmed it above
this height. Considering the PBL height may be as low as several hundreds to one thousand meters when
severe haze occurs in Jing–Jin–Ji (Wang et al., 2015a; Zhong et al., 2017), it may be concluded that the
whole PBL and its near upper atmosphere were cooled by aerosols to a varying extent during the different
stages of this haze process. The warming effects of aerosols above 750–850 hPa were very weak, and the
temperature differences among different days were also small. However, the cooling effects of aerosols
varied the most between different days from the surface to 975 hPa. For instance, surface daytime cooling
was about 2.2 K on 19 December, 1.5 K on 18 and 20 December, 1 K on 17 December, and 0.5–0.6 K on
15–16 December. This cooling effect of aerosols decreased rapidly with height. The difference in the
cooling rate between the surface and 850 hPa was 1.8 K on 19 December, 1.3 K on 18 and 20 December, 1
K on 17 December, and 0.3–0.4 K on 15 and 16 December. The difference in the cooling rate owing to
aerosols between the surface and the upper PBL was much bigger during the explosive growth stage than
the climbing stage. This may have resulted in further intensification of the temperature inversion layer that
already existed during the haze event, which will be discussed in the following section.

The meteorological data from the vertical soundings taken at Beijing and Xingtai were used to verify

this change in the temperature profile owing to aerosols. Figure 7 shows the vertical temperature profiles of
the sounding observations and the modeled temperature profiles of EXP1 and EXP2 during the climbing
stage (Figure 7a) and explosive growth stage (Figure 7b) at the two stations. The temperature profiles
(Figure 7a) show that the model results of EXP1 and EXP2 both simulated in part the observed temperature
inversion at Beijing and Xingtai on 15–16 December. The negligible difference between the temperature
profiles of EXP1 and EXP2 indicates that aerosol radiation had very little impact on the temperature
profiles and local inversion during the climbing stage. Nevertheless, Figure 7b shows that the observed
temperature inversions were obviously stronger and thicker on 18–19 December (explosive growth stage)
than those on 15–16 (climbing stage), both in Xingtai and Beijing. The temperate profiles of EXP2 were
much closer to the observational results than those of EXP1; and especially, the temperature inversions
were much stronger and also closer to observation than those of EXP1. This result proves that the
correction of local inversions by aerosols during the $PM_{2.5}$ explosive growth stage was effective.

However, it can also be seen that the inversions of EXP2, which included online AF, were still

weaker than observed at the two stations. This suggests there must be other reasons, besides the online
calculation of AF, for the underestimation of the observed extremely strong inversion by the model, which
is worthy of further study.

**3.5 Contributions of AF and decrease in turbulent diffusion coefficient to PM$_{2.5}$ explosive growth**

Turbulent diffusion is the main process of gas and particle exchange from surface to upper atmosphere, and removal by high-altitude transport, and one of the key tasks of atmospheric chemistry models is to capture this process. Firstly, the inversion and weak turbulent diffusion, which generates from atmospheric dynamic processes, leads to atmospheric stabilization and determines the occurrence of haze and its strength (Zheng et al., 2016). Once the haze occurs, aerosol radiation may in turn reinforce the inversion when aerosols exceed a certain critical value, leading to more PM$_{2.5}$ gathering near the ground. The relative importance of these two aspects on PM$_{2.5}$ explosive growth may vary with PM$_{2.5}$ concentrations and meteorological conditions, but they are irreplaceable for a reasonable prediction and simulation of PM$_{2.5}$ explosive growth and peaks in atmospheric models.

Figure 8 displays the hourly change in observed PM$_{2.5}$ (PM$_{2.5}$_OBS) and the modeled PM$_{2.5}$ of EXP1, EXP2 and EXP3, together with the modeled turbulent diffusion coefficient of the three experiments, in Beijing (Figure 8a) and Xingtai (Figure 8b), for the period 15–23 December. Comparison of the PM$_{2.5}$ modeled by EXP1, EXP2 and EXP3 with observation in Beijing (Figure 8a) shows that the PM$_{2.5}$ modeled by EXP3 was the closest to observation during the whole haze episode, which agreed with the results of the regional distribution of the explosive growth stage illustrated in Figure 4. EXP1 underestimated the PM$_{2.5}$ obviously during 17–22 December, and this underestimation was even more obvious with increasing PM$_{2.5}$. This difference between the modeled and observed PM$_{2.5}$ was largest during the explosive growth stage. AF reduced this difference to a considerable extent, and the PM$_{2.5}$ of EXP2 was much closer to observation than that of EXP1 during the explosive growth stage. However, there were certain differences between the observed and PM$_{2.5}$ and that modeled by EXP2, illustrating that AF cannot completely fill the sizeable gap between observed and modeled PM$_{2.5}$. The PM$_{2.5}$ of EXP3 reduced this gap further, showing the best agreement with observation, especially during the PM$_{2.5}$ explosive growth stage.

It can also be seen from Figure 8a that the diffusion coefficient of EXP1 was about 30–40 m$^2$/s during the explosive growth stage, which was about 50% of the 60–70 m$^2$/s on clear days (15 or 22 December). Obviously, this 50% diffusion coefficient difference between clear and severe haze days may be insufficient to separate the difference in turbulent diffusion intensity between the extremely stable atmosphere on haze

days and the unstable atmosphere on clear days, which is an important reason for the underestimated $PM_{2.5}$
explosive growth in EXP1. Compared with EXP1, the AF in EXP2 led to a notable enhancement of the
temperature inversion (Figure 7b), a significant decrease in the turbulent diffusion of $PM_{2.5}$ during the
explosive growth stage, and a low maximum diffusion coefficient at noon (as low as 14 $m^2$/s on 20
December – a reduction of 50% compared with EXP1). The maximum diffusion coefficient at noon on
haze days in EXP2 was only about 20% of that on clear days. The maximum diffusion coefficient at noon
in EXP3 was lower than 5 $m^2$/s on 20 December and, at the same time, the $PM_{2.5}$ modeled by EXP3 was
further increased and was also much closer to the observed $PM_{2.5}$ than the $PM_{2.5}$ of EXP2.

Through comparison of the temporal change of diffusion coefficient and $PM_{2.5}$ in EXP1, EXP2 and

EXP3 in Beijing, it is clear that an overestimation of turbulent diffusion coefficient owing to the absence of
online-calculated AF, as well as a deficient description of extremely stable stratification in the PBL scheme
of the atmospheric model, can lead to a distinct underestimation of $PM_{2.5}$ explosive growth and peaks when
severe haze occurs in China's Jing–Jin–Ji region.

The trends of change in diffusion coefficient and $PM_{2.5}$ at Xingtai in the three experiments (Figure 8b)

are similar to those at Beijing. The $PM_{2.5}$ of EXP3 was also closest to observation, followed by EXP2, and
then EXP1 was the worst, during the whole haze episode. However, during the explosive growth stage, the
relative contributions of AF and decrease in turbulent diffusion coefficient to the $PM_{2.5}$ peak values showed
some differences to those at Beijing. The contributions of decrease in turbulent diffusion coefficient to
$PM_{2.5}$ peaks were more important than those of AF at Xingtai. Located in the eastern foothills of the
Taihang Mountains, Xingtai is usually affected by downhill airflow. Temperature inversions in this area
form and strengthen easily, leading to stronger inversion, weaker turbulent diffusion, and more stable
atmospheric stratification. However, this kind of inversion and weak turbulent diffusion, derived from the
local terrain, is harder for PBL schemes in atmospheric chemistry models to describe, and likely
underestimated.

Figure 9 is a diagrammatic sketch of the contributions of AF and decrease in turbulent diffusion

coefficient to the $PM_{2.5}$ of the explosive growth stage according to the results at Beijing and Xingtai. It can
be seen that the diffusion coefficient of EXP1 was 30–35 $m^2$/s, while that of EXP2 was 15–17 $m^2$/s,
meaning AF reduced the diffusion coefficient by about 43%–57%, which led to the rise in simulated $PM_{2.5}$
from 144 ug/m$^3$ in EXP1 to 205 ug/m$^3$ in EXP2 at Beijing, and from 280 ug/m$^3$ in EXP1 to 360 ug/m$^3$ in
EXP2 at Xingtai. This means that AF reduced the underestimation of $PM_{2.5}$ at Beijing and Xingtai by 20%
and 25%, respectively. The diffusion coefficient of EXP3 was as low as 4–6 m$^2$/s during the explosive
growth stage, demonstrating the joint effects of AF and decrease in turbulent diffusion coefficient reduced
the diffusion coefficient to less than 4–6 m$^2$/s, near-zero, which we refer to as "turbulent intermittence".
The direct result of this "turbulent intermittence" was a further increase in the simulated surface $PM_{2.5}$,
Based on EXP2, the further decrease in turbulent diffusion coefficient reduced the underestimation of
simulated $PM_{2.5}$ by 14% to 20%, and the $PM_{2.5}$ errors in EXP3 were reduced to as low as −11% to 2%.
**4. Conclusions**
Using an atmospheric chemistry model, GRAPES_CUACE, three experiments (EXP1, EXP2 and
EXP3) were designed to study the reason for the explosive growth of $PM_{2.5}$ mass during a "red-alert" heavy
haze event that occurred during 15–23 December 2016 in China's Jing–Jin–Ji region. The contributions of
AF and decrease in turbulent diffusion coefficient to the $PM_{2.5}$, representing compensation for the deficient
description of extremely weak turbulent diffusion in the PBL scheme of the atmospheric model, were
studied by analyzing the changes in $PM_{2.5}$, SDSRF, wind speed and temperature, diffusion coefficient, and
the relationships among them, in the three experiments.
Results show that the diffusion coefficient in EXP1 was about 60–70 m$^2$/s on clear days and 30–35
m$^2$/s on haze days. The 50% difference between the two was considered insufficient to separate the unstable
atmosphere on clear days and the extreme stable atmosphere on severe haze days, compared with the
differences in direct downward solar radiation between clear and haze days, which was also proven
indirectly by the weaker inversion of EXP1 than that from sounding observations. This led to a 40%–51%
underestimation of the $PM_{2.5}$ peaks in EXP1 during the $PM_{2.5}$ explosive growth stage. Online calculation of
AF reduced the surface and PBL wind speed and cooled the surface and PBL atmosphere. The surface
daytime cooling due to aerosol radiation was 1.5–2.2 K during the explosive growth stage and 0.5–0.6 K
during the climbing stage. The cooling effect of aerosols decreased rapidly with height, and this was a
major reason for the strengthening of the temperature inversion during the explosive growth stage. The
reduced diffusion coefficient owing to AF reached 43%–57% during the $PM_{2.5}$ explosive growth stage. The
local inversion simulated in EXP2 was strengthened and closer to the actual sounding observation than that
of EXP1. This resulted in a 20%–25% reduction in the underestimation of $PM_{2.5}$, with $PM_{2.5}$ errors in EXP2
being as low as −16 to −11% during the explosive growth stage. The impact on $PM_{2.5}$ owing to AF in the
model run was distinct during the explosive growth stage, but minor during the climbing stage, indicating a
critical value of 150 ug/m$^3$ of $PM_{2.5}$ leading to an effective AF in online atmospheric chemistry models.
However, the local inversion simulated by EXP2 was still weaker than observed, and the $PM_{2.5}$ of EXP2
was still smaller than observed, illustrating AF could not solve all the $PM_{2.5}$ underestimation problems. In
EXP3, the decrease in turbulent diffusion coefficient of particles and gas based on EXP2 resulted in a
14%–20% lessening of the $PM_{2.5}$ underestimation based on EXP2, and the $PM_{2.5}$ errors of EXP3 were
reduced to −11% to 2%.
The present study illustrates that the PBL schemes in current atmospheric chemistry models are
probably insufficient for describing the extremely stable atmosphere resulting in explosive growth of $PM_{2.5}$
and severe haze in China's Jing–Jin–Ji region. This may involve two important reasons: the absence of an
online calculation of AF, and/or a deficient description of extremely weak turbulent diffusion by the PBL
scheme in the atmospheric chemistry model. Our study suggests that an online calculation of AF and an
improvement in the representation of turbulent diffusion in PBL schemes, with a focus on extremely stable
atmospheric stratification, in atmospheric chemistry models, are indispensable for a reasonable description
of local "turbulent intermittence" and an accurate prediction of the explosive growth and peaks of $PM_{2.5}$ of
severe haze in China's Jing–Jin–Ji region.


**Author Contributions:**
Hong Wang and Xiaoye Zhang designed the idea and experiments; Hong Wang and Yue Peng carried them
out; Hongli Liu prepared the emissions data and introduction; Meng Zhang performed some of the model
runs; Huizheng Che and Yu Zheng processed the AOD and SSA observational data; Yanli Cheng completed
Table 3 and the related introduction.
**Acknowledgements**
This study was supported by the National Key Project (2016YFC0203306), the National Natural Science
Foundation of China (41590874), and the National (Key) Basic Research and Development (973) Program
of China (2014CB441201).

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

Table 1. Physical and chemical processes in GRAPES_CUACE

| Process | Option | Reference |
|---|---|---|
| Explicit precipitation | WDM6 | Lim and Hong (2010) |
| Cumulus cloud | KFETA scheme | Kain (2004) |
| Longwave radiation | Goddard | Chou et al. (2001) |
| Shortwave radiation | Goddard | Chou et al. (1998) |
| Surface layer | SFCLAY scheme | Pleim (2007) |
| PBL | MRF scheme | Hong et al. (1996, 2006) |
| Land surface | SLAB scheme | Kusaka et al. (2001) |
| Gas-phase chemistry | RADM II | Stockwell et al. (1990) |
| Aerosol | CUACE | Zhou et al. (2012) |
| Aerosol direct effect | External mixing | Wang et al. (2015) |
| Aerosol indirect effect | CAUCE+WDM6 | Zhou et al. (2016) |

















Table 2. Design of sensitivity experiments



| Experiment | Description |
|---|---|
| EXP1 | Background experiment: ignoring aerosol radiation and conventional  diffusion coefficient   of chemical tracers by PBL scheme in GRAPES_CUACE |
| EXP2 | Online AF online and conventional   diffusion coefficient   of chemical tracers by PBL scheme in GRAPES_CUACE |
| EXP3 | Online AF and DC of chemical tracers set to 20% of conventional  diffusion coefficient   calculated by PBL scheme, representing compensation for the deficient description of extremely weak turbulent diffusion by the PBL scheme; diffusion coefficient   in physical and dynamic processes the same as EXP1 |


Table 3. VOCs in the emissions data

| VOC | Full name |
|---|---|
| ALD | Acetaldehyde and higher aldehydes |
| CH4 | Methane |
| CSL | Cresol and other hydroxy substituted aromatics |
| ETH | Ethane |
| HC3 | Alkanes w/ $2.7 \times 10^{-13} >$ kOH $< 3.4 \times 10^{-12}$ |
| HC5 | Alkanes w/ $3.4 \times 10^{-12} >$ kOH $< 6.8 \times 10^{-12}$ |
| HC7 | w/kOH $> 6.8 \times 10^{-12}$ |
| HCHO | Formaldehyde |
| ISOP | Isoprene |
| KET | Ketones |
| OL2 | Ethene |
| OLI | Internal olefins |
| OLT | Terminal olefins |
| ORA2 | Acetic and higher acids |
| PAR | Paraffin carbon bond |
| TERPB | Monoterpenes |
| TOL | Toluene and less reactive aromatics |
| XYL | Xylene and more reactive aromatics |

Table 4. Observed and modeled daily AOD (* stands for shortage of observation)

| Date | Shijiazhuang | | Beijing | | Xianghe | |
|------|------|-------|------|-------|------|-------|
| | OBS | MODEL | OBS | MODEL | OBS | MODEL |
| 15 | 0.46 | 0.55 | 0.07 | 0.12 | 0.10 | 0.15 |
| 16 | 0.62 | 0.60 | 0.14 | 0.18 | 0.60 | 0.40 |
| 17 | 1.30 | 1.10 | 0.50 | 0.56 | 1.33 | 1.05 |
| 18 | 1.42 | 1.20 | 0.69 | 0.75 | 0.87 | 0.97 |
| 19 | 1.26 | 1.30 | 0.50 | 0.86 | 0.96 | 0.90 |
| 20 | * | 1.20 | 1.90 | 1.70 | * | 1.50 |
| 21 | * | 0.65 | 1.76 | 1.50 | 1.78 | 1.60 |
| 22 | 0.18 | 0.30 | 0.10 | 0.20 | 0.18 | 0.22 |

Table 5. Observed and modeled daily SSA (* stands for shortage of observation).

| Date | Shijiazhuang | | Beijing | | Xianghe | |
|------|------|-------|------|-------|------|-------|
| | OBS | MODEL | OBS | MODEL | OBS | MODEL |
| 15 | 0.83 | 0.85 | 0.81 | 0.83 | 0.86 | 0.84 |
| 16 | 0.83 | 0.85 | 0.88 | 0.86 | 0.92 | 0.86 |
| 17 | 0.88 | 0.89 | 0.88 | 0.90 | 0.93 | 0.90 |
| 18 | 0.87 | 0.89 | 0.91 | 0.92 | 0.90 | 0.90 |
| 19 | 0.86 | 0.91 | 0.90 | 0.93 | 0.92 | 0.91 |
| 20 | * | 0.90 | * | 0.93 | * | 0.92 |
| 21 | * | 0.88 | 0.93 | 0.93 | * | 0.90 |
| 22 | 0.82 | 0.83 | 0.84 | 0.86 | 0.88 | 0.84 |


**Figure captions**

**Fig. 1.** (a) Model domain and location of Jing–Jin–Ji. (b) Geographic location and topography of Jing–Jin–
Ji. Blue dots are the locations of $PM_{2.5}$ observations; red triangles are the locations of automatic weather
stations; yellow stars are the two sounding stations; black crosses are the CARSNET and AEROSNET
stations.
**Fig. 2.** Geopotential height (color-shaded; gp10m), temperature (dashed black contours; K) and wind (wind
bars; m/s) in the (a) upper (500 hPa) and (b) middle (700 hPa) atmosphere, and geopotential height and
wind in the (c) lower atmosphere (850 hPa) and (d–f) PBL (900, 950, 1000 hPa), at 0000 UTC 19
December 2016.
**Fig. 3.** Observed and modeled wind speed and temperature at the surface (upper panels), and the PBL-mean
wind speed and temperature (lower panels), from the results of EXP1, EXP2 and EXP3 for Beijing, Xingtai,
and the average for Jing–Jin–Ji as a whole, during 15–24 December 2016.
**Fig. 4.** Mean observed ($OBS\_PM_{2.5}$) and modeled $PM_{2.5}$ concentration ($\mu g/m^3$) of the $PM_{2.5}$ explosive
growth stage, from the results of EXP1, EXP2 and EXP3 ($PM_{2.5}\_EXP1$, $PM_{2.5}\_EXP2$ and $PM_{2.5}\_$ EXP3,
respectively).
**Fig. 5.** Mean percentage change in SDSRF ($W/m^2$) owing to (a) aerosols and (b) aerosols+ decrease in
turbulent diffusion coefficient during the explosive growth stage.
**Fig. 6.** Profiles of average temperature change in Jing–Jin–Ji owing to AF (K) during 15–20 December

2016.

**Fig. 7.** Sounding-observed and modeled temperature profiles in EXP1 and EXP2 during the (a) climbing
stage and (b) explosive growth stage in Beijing and Xingtai.
**Fig. 8.** Hourly change of $PM_{2.5}\_OBS$, $PM_{2.5}\_EXP1$, $PM_{2.5}\_EXP2$, and $PM_{2.5}\_EXP3$ ($\mu g/m^3$), together with
the diffusion coefficient (DC) at 950 hPa of the three experiments (DC_EXP1, DC_EXP2, DC_EXP3)
during 15–22 December 2016 in (a) Beijing and (b) Xingtai.
**Fig. 9.** Diagrammatic sketch of the contributions of AF and decrease in turbulent diffusion coefficient
(DTD) to the $PM_{2.5}$ explosive growth.


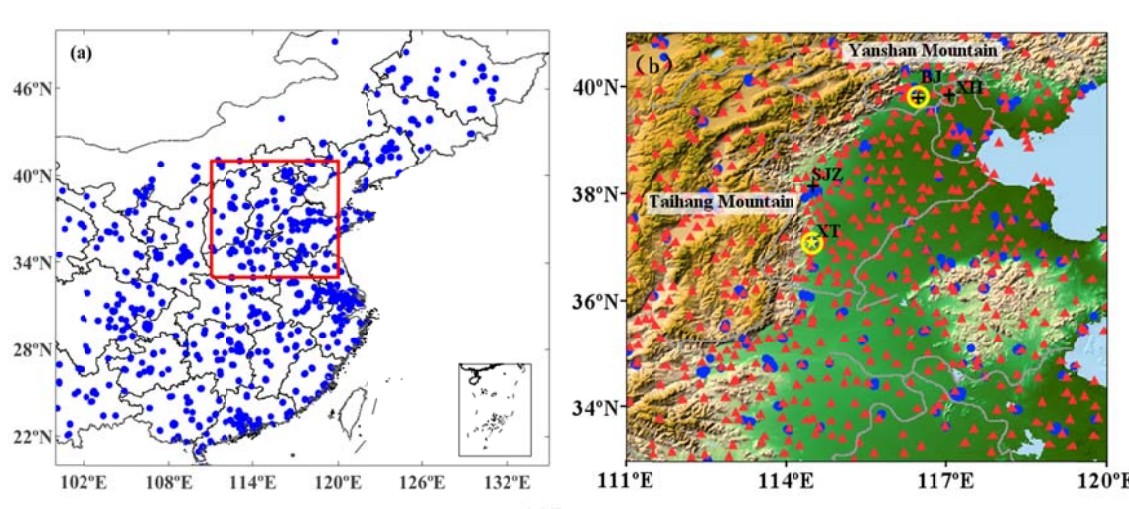

**Fig. 1.** (a) Model domain and location of Jing–Jin–Ji. (b) Geographic location and topography of Jing–Jin–
Ji. Blue dots are the locations of PM$_{2.5}$ observations; red triangles are the locations of automatic weather
stations; yellow stars are the two sounding stations; black crosses are the CARSNET and AEROSNET
stations.

775
776

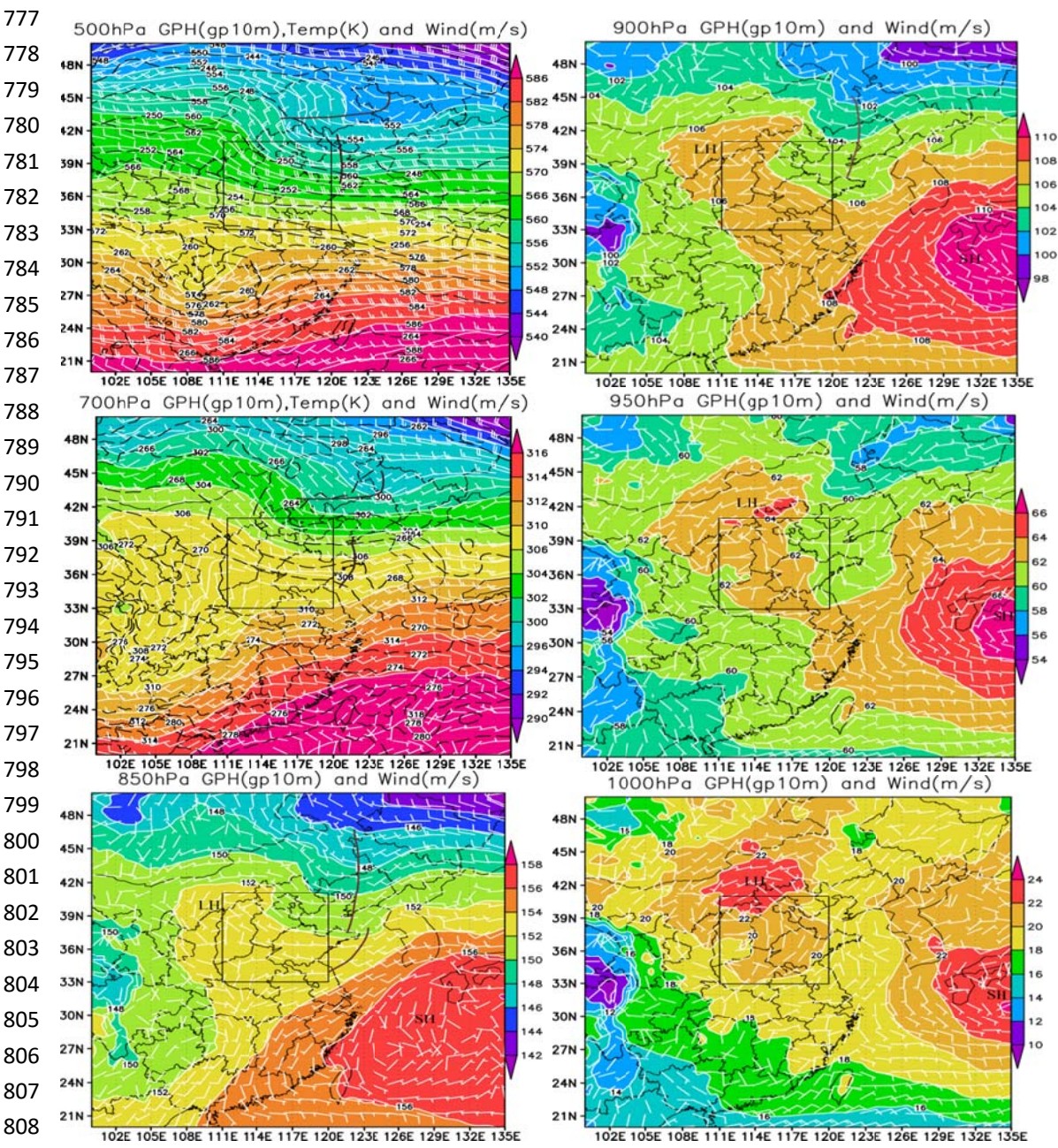

**Fig. 2.** Geopotential height (color-shaded; gp10m), temperature (dashed black contours; K) and wind (wind bars; m/s) in the (a) upper (500 hPa) and (b) middle (700 hPa) atmosphere, and geopotential height and wind in the (c) lower atmosphere (850 hPa) and (d–f) PBL (900, 950, 1000 hPa), at 0000 UTC 19 December 2016.


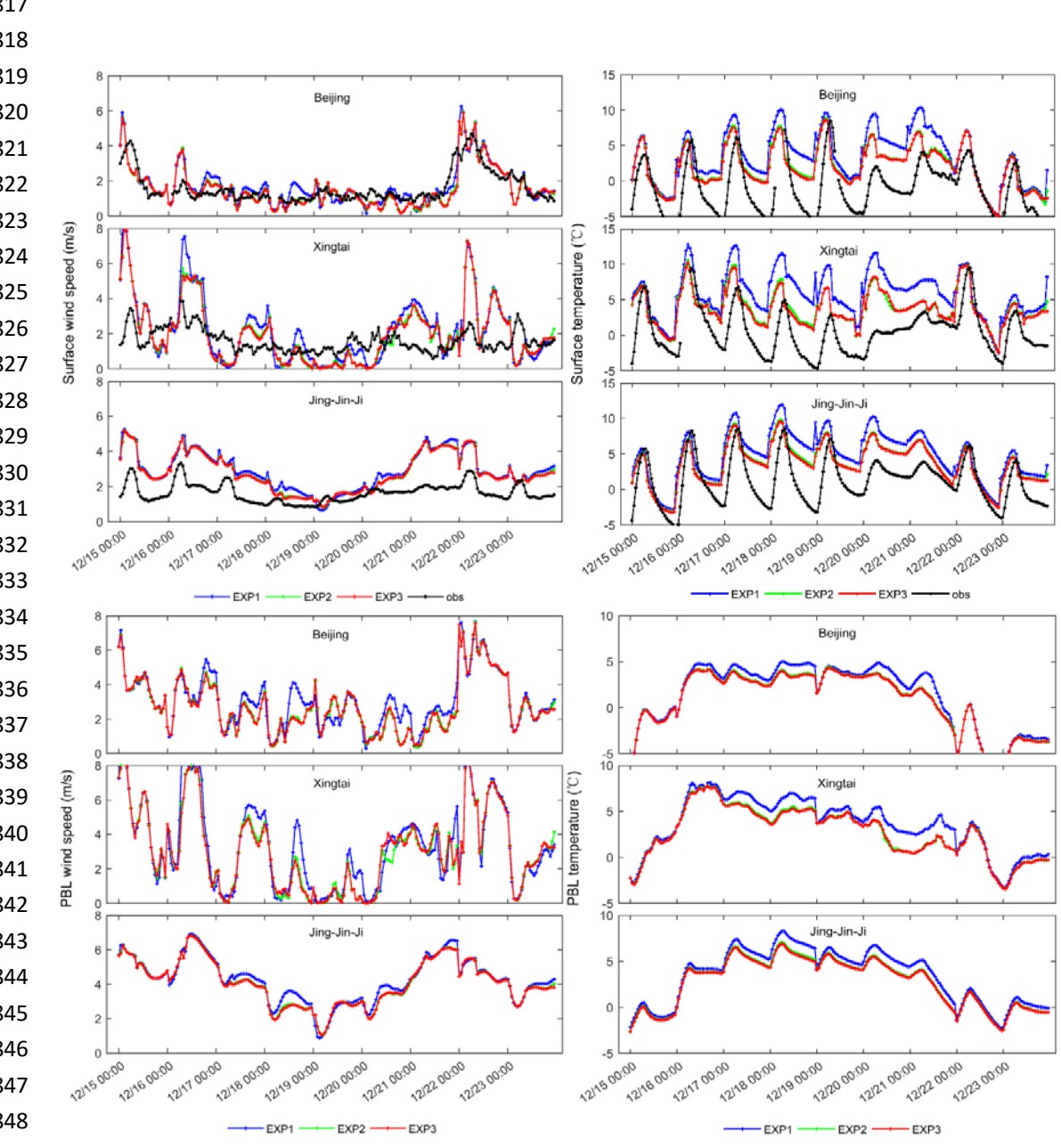

**Fig. 3.** Observed and modeled wind speed and temperature at the surface (upper panels), and the PBL-mean
wind speed and temperature (lower panels), from the results of EXP1, EXP2 and EXP3 for Beijing, Xingtai,
and the average for Jing–Jin–Ji as a whole, during 15–24 December 2016.




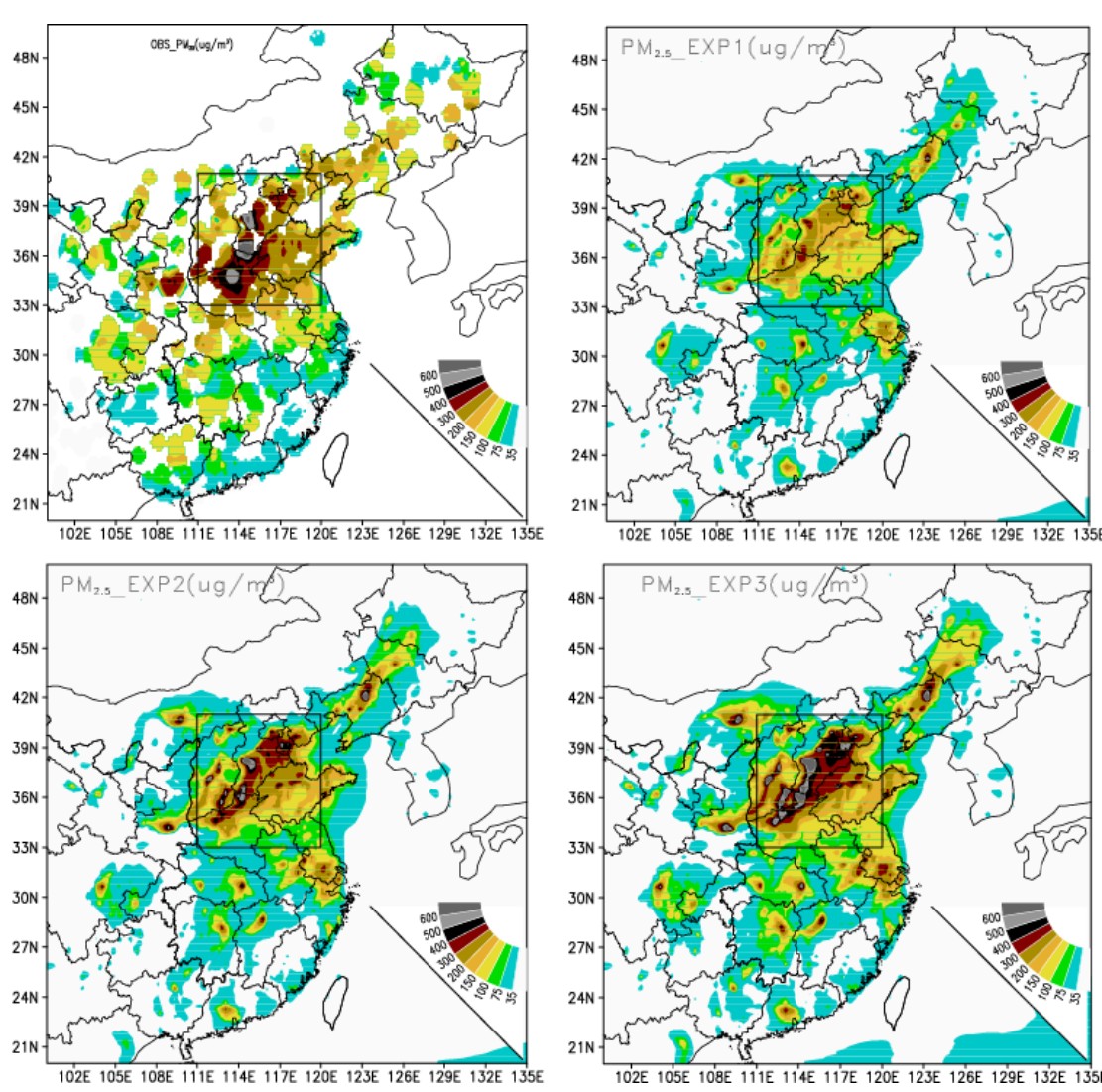



**Fig. 4.** Mean observed (OBS_PM$_{2.5}$) and modeled PM$_{2.5}$ concentration (μg/m$^3$) of the PM$_{2.5}$ explosive
growth stage, from the results of EXP1, EXP2 and EXP3 (PM$_{2.5}$_EXP1, PM$_{2.5}$_EXP2 and PM$_{2.5}$_ EXP3,
respectively).






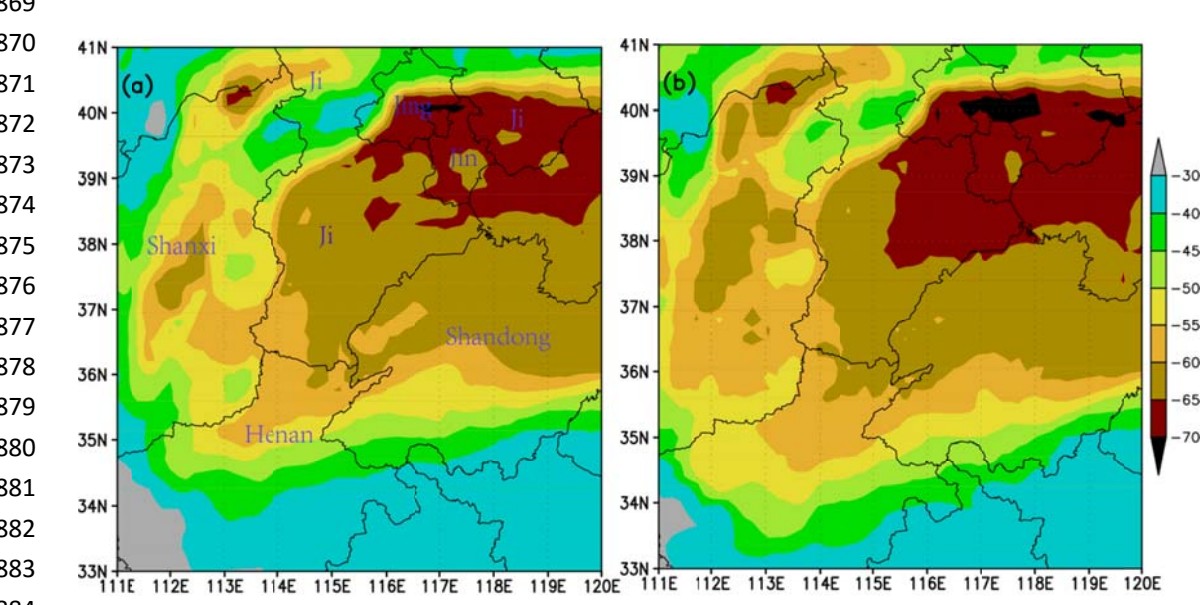

**Fig. 5.** Mean percentage change in SDSRF (W/m$^2$) owing to (a) aerosols and (b) aerosols+ decrease in

turbulent diffusion coefficient during the explosive growth stage.





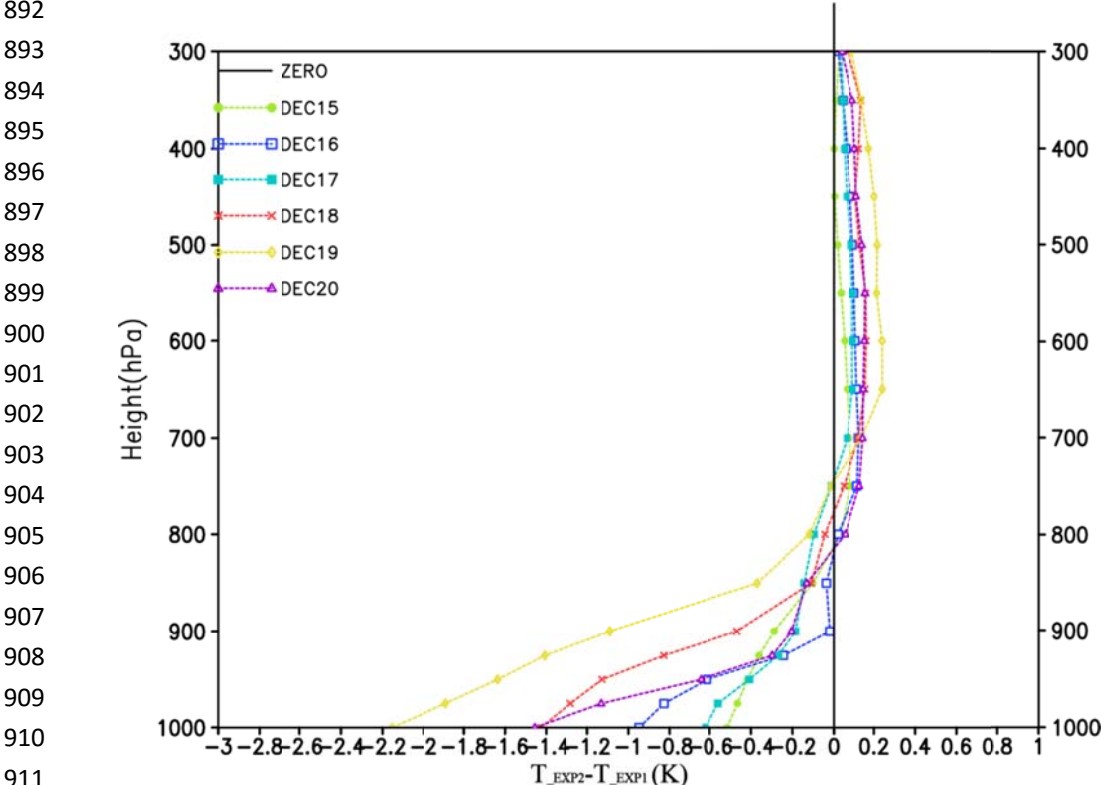

**Fig. 6.** Profiles of average temperature change in Jing–Jin–Ji owing to AF (K) during 15–20 December 2016.

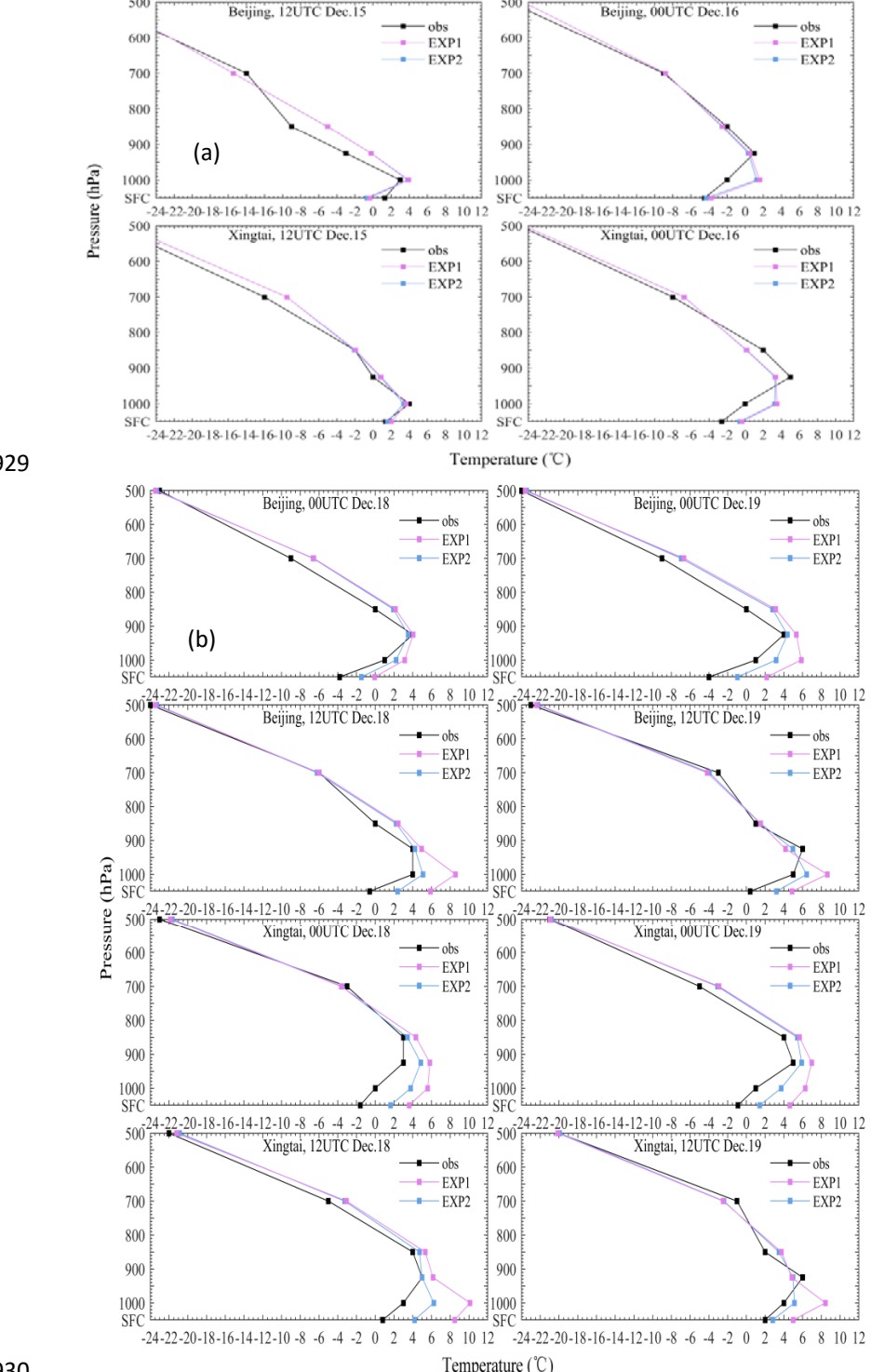

**Fig. 7.** Sounding-observed and modeled temperature profiles in EXP1 and EXP2 during the (a) climbing

stage and (b) explosive growth stage in Beijing and Xingtai.

933

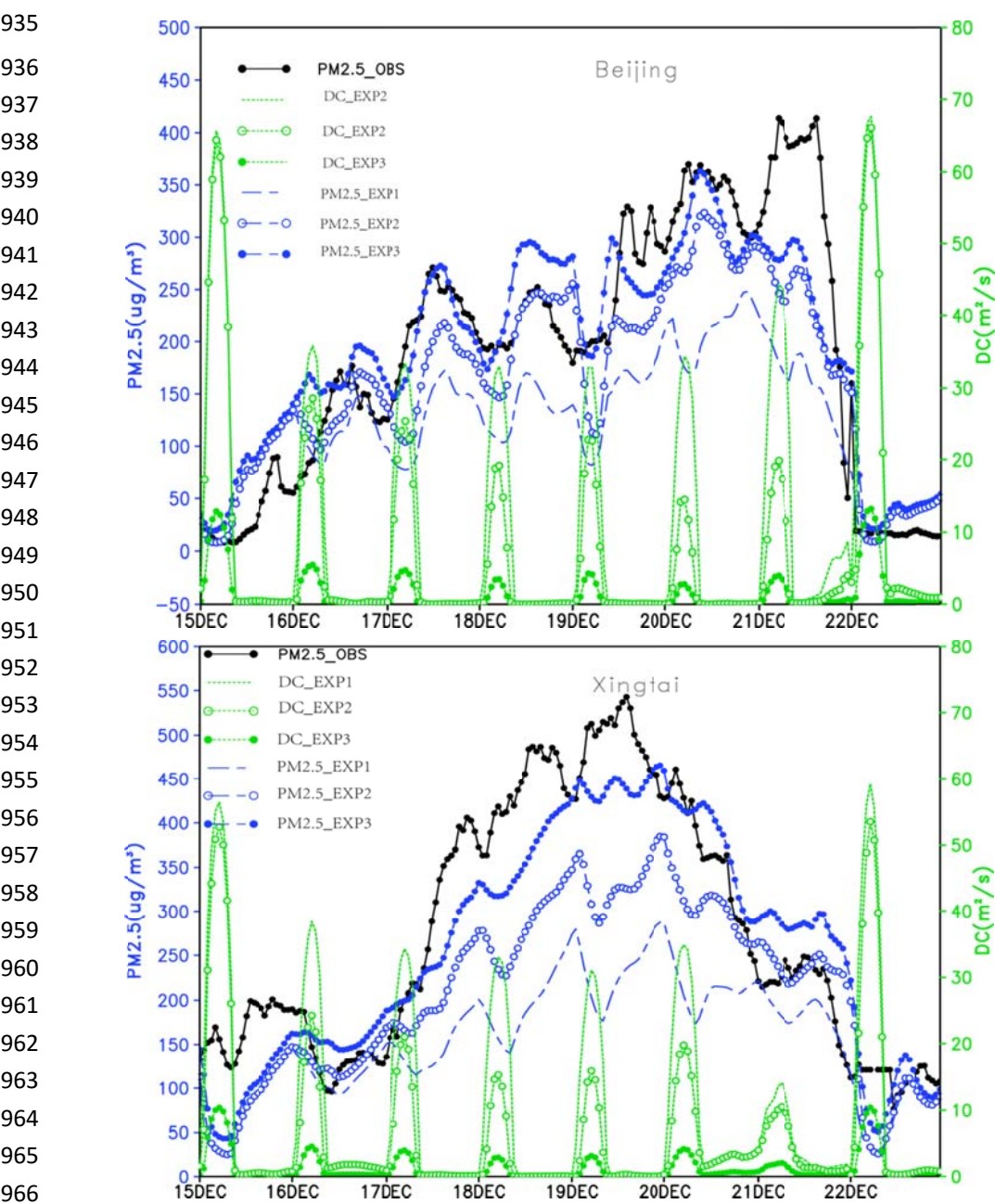

**Fig. 8.** Hourly change of PM$_{2.5}$_OBS, PM$_{2.5}$_EXP1, PM$_{2.5}$_EXP2, and PM$_{2.5}$_EXP3 (μg/m$^3$), together with

the diffusion coefficient (DC) at 950 hPa of the three experiments (DC_EXP1, DC_EXP2, DC_EXP3)

during 15–22 December 2016 in (a) Beijing and (b) Xingtai.

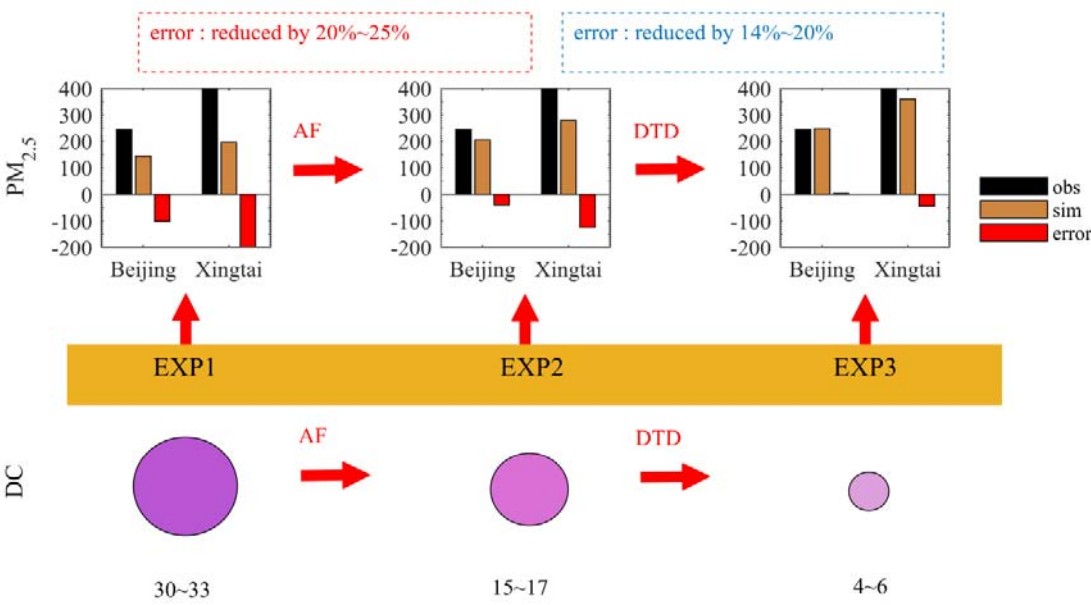

**Fig. 9.** Diagrammatic sketch of the contributions of AF and decrease in turbulent diffusion coefficient
(DTD) to the $PM_{2.5}$ explosive growth.