# Peer review of "The Contributions to the Explosive Growth of $PM_{2.5}$ Mass due to Aerosols-Radiation Feedback and Further Decrease in Turbulent Diffusion during a Red-alert Heavy Haze in Jing-Jin-Ji in China"

_Atmospheric Chemistry and Physics, 2018_

## Referee Comment (RC1) · Anonymous Referee #1 · 22 Jul 2018

The understanding of atmospheric boundary layer and its impact on air quality is an important issue in atmospheric environment study. Focusing this scientific issue, this paper investigated the effect of aerosols-radiation feedback on turbulent diffusion during a Red-alert Heavy Haze in JING-JIN-JI in China, by employing the atmospheric chemical model GRPAES_CUACE with three simulation experiments. It is interesting to investigate the impacts of aerosols-radiation feedback on PM2.5 changes between the climbing stage and explosive growth stage. This study results illustrated that the

[Figure]

PBL scheme in current atmospheric chemical models is probably insufficient for describing the extremely stable atmosphere in explosive growth of PM2.5 during severe haze events in JING-JIN-JI in China, which may involve in two important reasons: One is the absence of online calculation of AF, another is the deficient description of the extreme weak turbulent diffusion in the PBL scheme in the atmospheric chemical model. This manuscript presenting the interesting results could improve our understanding on environment changes and fall within the scope of ACP. I suggest the minor revisions before it is published as follows:

1. The paper needs to give the model settings of GRPAES_CUACE, such as physical and chemical parameterizations.

2. It needs to add meteorological factors evaluation, especially wind speed, because wind speed has a deeply influence on diffusion of PM2.5, and temperature inversion in PBL.

3. It could be better to add turbulent diffusion coefficients calculated by observation data if possible.

4. Please compare the downward long radiation in three experiments to figure out the contribution of aerosols.

---

## Referee Comment (RC2) · Anonymous Referee #2 · 31 Jul 2018

This paper investigated the impact of aerosol radiation feedback and decreased turbulent diffusion on PM2.5 during a heavy polluted episode in China. The objectives of this research might be interesting and potentially important; however, I have a number of concerns with the manuscript.

General comments

First, the lack of description about the GRPAES_CUACE model is troubling. What

are the basic physical parameterizing schemes and chemical mechanism used in this study? How the model treat those crucial processes, such as SOA formation, two-way coupling, BC mixing states, aging processes. More important, how the model calculate the diffusion mixing? Any deficiency that can explain the supposed underestimation in diffusion coefficient, beside the lack of the aerosol radiative effect?

Second, I suggest the authors to provide additional validation of the model performance. How was the model performance in simulating the meteorological variables, PM chemical components and precursors? Does the underestimation apply to all PM components? It is also very important to exam that how the change in diffusion influence on the model performance in simulating species including both PM chemical components and precursor, since the mixing process is critical in determining the concentrations of all species.

Third, the description about scenario design need be elaborated. In EXP_td_af, how the dynamic field is updated by the aerosol feedback, and is there any nudging processed? In EXP_td20_af, how was the 80% reduction in turbulent diffusion implemented in the model. Did the change apply to all simulated domains? Is there any evidences or references which can support such modification? Based on the results (overestimation is found for clean days and areas outside JJJ), I don't think the DTD is applicable for all grid cells and days and can explain the underestimation of PM2.5.

Specific comments

Title: need provide some description about "Red-alert" in introduction section

Line 83: "GRAPES_CUACE", provide the full name and some references about the model.

Line 89: How to get the boundary conditions?

Line 92: "The model horizontal resolution is adopted as $0.15°\times0.15°$". Is it high enough to capture the strong inversion during the episode? What about the vertical resolution?

Line 100: I would suggest the authors to elaborate the section 2.2. Is the emission data open to the public? What's the accuracy of the data? How does it compare to the others inventories, such as MEIC, EDGAR, etc? How was the spatial / temporal allocation processed?

Line 101: "human life", is it "domestic"?

Line 105-106: need provide full names for the VOC species

Line 121: "a further 80% decrease in turbulent diffusion (DTD) of chemical tracers based on EXP_td_af representing a compensation for the insufficient description of extremely weak turbulent diffusion by PBL scheme in atmospheric chemical model". how the 80% decreased DTD was determined? Was the overestimation of vertical mixing is due to the coarse resolution, or underestimation of aerosol feedback?

Line 134: in section 3.1, what about PM chemical component? The mixing basically can revolve the total PM mass. However, if the chemical profile doesn't agree well the observation, it still cannot solve the issue.

Line 155: "Some studies offline and online", is it "some offline/online modeling studies"?

Line 157: "AF of composite aerosols from black carbon, organic carbon, sulfate, nitrate, dust, ammonium, and sea salt aerosols had been online coupled into the in GRAPES_CAUCE model." how does the model treat mixing states and aging process? How is the model performance in simulating the PM components and AF?

Line 173: "the temperature inversion layer pre-existed during the haze event", it is not easy to see the temperature inversion in the plots.

Line 182: "Figure 4b shows that the observed temperature inversions were obvious stronger and the inversion depth thicker on 18 to 19 (during EGS of PM2.5) than those on 15 to 16 Dec (CS of PM2.5" But the PBL height seems opposite, lower on 18 to 19 but higher on 15 to 16 Dec.

Line 191: "The contributions to PM2.5 EG due to AF and DTD". Since AF also contributes to DTD, how to separate these two effects.

Line 207: "Exp_bk under underestimated the PM2.5", "under" should be deleted

Line 224: "the overestimation of turbulent DC", is there any observation data to prove the overestimation of DC?

Figure 2: The PM2.5 in area outside JJJ seems all overestimated. The af/td cases make it even worse. Seems like it is not proper to apply the 80% DTD to all grid cells.

Figure 3: please clarify that the data is regional average in JJJ.

Figure 4: what about the days when PM reach peak for Dec 20-22 in Beijing.

Figure 5: PM2.5_td_af seems more reasonable than PM2.5_td20_af, in consideration of the possible missing heterogeneous chemistry. What's the reason for the underestimation of the peak on Dec 21, even though the DC is already very low.

Figure 6: the figure is misleading. Since the reduced error in td20_af is because that the overestimation on Dec 18 compensates the underestimation on Dec 21 in Beijing.

---

## Referee Comment (RC3) · Anonymous Referee #3 · 1 Aug 2018

This paper deals with the effect of "aerosols-radiation feedback" and "decrease in tur-bulent diffusion" to "the Explosive growth of PM2.5 mass" in Jing-Jin-Ji area, northern China. Numerical experiments are carried out for three runs, the first run absents "Aerosols-Radiation Feedback", the second run is with normal Aerosols-Radiation Feedback, and the third run is with reduced Turbulent Diffusion in addition. A one-week haze event is modeled. Results of these runs, one by one, show improvement to reproduce the observed results.

[Figure]

My major concern and suggestion: 1)This paper proposes a sensitive test on factors that influence the model result. But in the paper, results are directly presented, no middle results or any more supporting materials. Therefore, the conclusions are not convinced. 2)Reducing DC may lead the meteorological model running unrealistically. Details about the change of wind field etc. need to be displayed. 3)Need description: synoptic background/weather condition for this haze event. 4)Details of the model are needed, particularly the parts of lower atmosphere, levels, PBL scheme, surface model, radiation, aerosol absorption, etc. 5)PBL is mentioned as an crucial part in the paper, but no information about PBL is illustrated.

Other points: 1)"Jing-Jin-Ji", not to be "JING-Jin-Ji" etc. different forms. 2)Too many abbreviates, and their combination, hard to read the text; There are only 3 experiment runs, number them as Run 1∼3, may be clearer. 3)Page 4, line 70-72:"One is that aerosols radiation feedback (AF) is not calculated online in the model run. AF can restrain turbulence by cooling surface and PBL while heating the atmosphere above it", Result of AF is mostly determined by absorbing aerosols, and by their vertical distribution. 4)Page 4, line 77: "A Red-alert Heavy Haze occurred on 15 to 17 December", 15-23 Dec. 5)Page 4, Section 2.1, the model GRAPES_CUACE need to be introduced more detail, as well the setup of the simulations. 6)Page 5, Section 2.2, just lists the air pollutants, not relevant information crucial to this paper is given. 7)Page 5-6, Section 2.4, too simple in description. Table 1, repeated, but still too simple. 8)Page 6, line 131: "which is named as the explosive growth (EG)", this is the first time mentions "explosive growth". Nothing is known what is the cause of EG: chemistry, transport, or accumulation of air pollutant? 9)Page 6, Section 3.1, only PM2.5 is investigated. What about its source: primary or secondary? What about other pollutants? And their effect on PM2.5 concentration? 10)Page 7, Section 3.2, directly presents result of temperature profile, no logic description about the relation of AF and inversion strengthening. No qualitative and quantitative assessment on question if the result is right or correct. 11)Page 8, Section 3.3, the text is very difficult to read through since too many abbreviates. 12)Page 9, line 220-221: " significant decrease in turbulent diffusion on PM2.5 during

EGS and DC_td_af was as low as 14m2/s on 20 December, which decreased about 50% comparing with DC_bk.", this sentence need to clarify. And "DC was 14m2/s", in where? What level? What time? Day or night? 12)Page 10, line 245: "...we name it as 'turbulent intermittent'", What do you mean the 'turbulent intermittent'? Does 'turbulent intermittent' really mean lower diffusion coefficient or mixing rate? 13)Page 10, line 253-254: "for the deficient description of extreme weak turbulent diffusion by PBL scheme in atmospheric models, are studied by analysing the changes of...", nothing about the PBL scheme is presented in this paper. 14) in Table 1, "retaining 20% (reducing 80%) of normal turbulent diffusion", How to do this? Reducing the value at all the model domain? 15)in Figure 5, the DC, at what position? What level/height?

---

## Author Comment (AC1) · 19 Sep 2018

Anonymous Referee #1 The understanding of atmospheric boundary layer and its impact on air quality is an important issue in atmo-

spheric environment study. Focusing this scientific issue, this paper investigated the effect of aerosols-radiation feedback on turbulent diffusion during a Red-alert Heavy Haze in JING-JIN-JI in China, by employing the atmospheric chemical model GR-PAES_CUACE with three simulation experiments. It is interesting to investigate the impacts of aerosols-radiation feedback on PM2.5 changes between the climbing stage and explosive growth stage. This study results illustrated that the PBL scheme in current atmospheric chemical models is probably insufficient for describing the extremely stable atmosphere in explosive growth of PM2.5 during severe haze events in JING-JIN-JI in China, which may involve in two important reasons: One is the absence of online calculation of AF, another is the deficient description of the extreme weak turbulent diffusion in the PBL scheme in the atmospheric chemical model. This manuscript presenting the interesting results could improve our understanding on environment changes and fall within the scope of ACP. I suggest the minor revisions before it is published as follows:

Response: We would like to heartily thank the reviewer for his serious review on our work and the valuable comments. We carefully considered comments of the reviewer and revised the paper accordingly, one by one of the following:

Comment 1 The paper needs to give the model settings of GRPAES_CUACE, such as physical and chemical parameterizations.

Response: The model settings including dynamic frame, physical and chemical parameterizations is summarized in Table 1 and the related text is rewritten in line 92-128 in section 2.1 in the revised manuscript.

Comment 2. It needs to add meteorological factors evaluation, especially wind speed, because wind speed has a deeply influence on diffusion of PM2.5, and temperature inversion in PBL.

Response: Wind speed evaluation and study (figure 3) are added in the revised manuscript; Geopotential height and temperature are also offered in figure 2 in the

revised manuscript. The related text is also added in the manuscript.

Comment 3. It could be better to add turbulent diffusion coefficients calculated by observation data if possible.

Response: YesïijŇit is better if the turbulent diffusion coefficients based on observation data is calculated and compared with simulated ones. This need the daytime observation data of vertical profiles of PBL meteorology including wind, potential temperature, and PBL height ect. Unfortunately, the sounding meteorology data in the study area are at 00 UTC(early morning in local time) and 12 UTCïijĹdusk in local timeïijĽ, so it is very difficult to add turbulent diffusion coefficients calculated by observation data at present.

Comment 4. Please compare the downward long radiation in three experiments to figure out the contribution of aerosols.

Response: The downward shortwave radiation fluxes due to AR and DTD (figure 5 and the related text) are added in the revised manuscript according to the reviewer's comment.

Please also note the supplement to this comment:
https://www.atmos-chem-phys-discuss.net/acp-2018-512/acp-2018-512-AC1-supplement.pdf

---

## Author Response (AR1)

The understanding of atmospheric boundary layer and its impact on air quality is an important issue in atmospheric environment study. Focusing this scientific issue, this paper investigated the effect of aerosols-radiation feedback on turbulent diffusion during a Red-alert Heavy Haze in JING-JIN-JI in China, by employing the atmospheric chemical model GRPAES_CUACE with three simulation experiments. It is interesting to investigate the impacts of aerosols-radiation feedback on PM2.5 changes between the climbing stage and explosive growth stage. This study results illustrated that the PBL scheme in current atmospheric chemical models is probably insufficient for describing the extremely stable atmosphere in explosive growth of PM2.5 during severe haze events in JING-JIN-JI in China, which may involve in two important reasons: One is the absence of online calculation of AF, another is the deficient description of the extreme weak turbulent diffusion in the PBL scheme in the atmospheric chemical model. This manuscript presenting the interesting results could improve our understanding on environment changes and fall within the scope of ACP. I suggest the minor revisions before it is published as follows:

**Response:**

We would like to heartily thank the reviewer for his serious review on our work and the valuable comments. We carefully considered comments of the reviewer and revised the paper accordingly, one by one of the following:

**Comment 1** The paper needs to give the model settings of GRPAES_CUACE, such as physical and chemical parameterizations.

**Response:**

The model settings including dynamic frame, physical and chemical parameterizations is summarized in Table 1 and the related text is rewritten in line 92-128 in section 2.1 in the revised manuscript.

**Comment 2**. It needs to add meteorological factors evaluation, especially wind speed, because wind speed has a deeply influence on diffusion of PM$_{2.5}$, and temperature inversion in PBL.

**Response:**

Surface and PBL wind speed and temperature evaluation and study (figure 3 and the related text in line 227-257) are added in the revised manuscript; AOD (Table 4) and SSA (Table 5)

evaluation (text in line 258-269) are also added in the revised manuscript.

**Comment 3**. It could be better to add turbulent diffusion coefficients calculated by observation data if possible.

**Response:**

Yes,it is better if the turbulent diffusion coefficients based on observation data is calculated and compared with simulated ones. This need the daytime observation data of vertical profiles of PBL meteorology including wind, potential temperature, and PBL height ect. Unfortunately, the sounding meteorology data in the study area are at 00 UTC(early morning in local time) and 12 UTC(dusk in local time), so it is very difficult to add turbulent diffusion coefficients calculated by observation data at present.

**Comment 4.** Please compare the downward long radiation in three experiments to figure out the contribution of aerosols.

**Response:**

Figure 5, and the related discussion section "**3.3 The downward solar radiation flux change by aerosols and DTD experiment**" in line 291-312 are added to discuss the downward shortwave radiation fluxes due to AR and DTD in the revised manuscript according to the reviewer's comment.

**Anonymous Referee #3**

This paper deals with the effect of "aerosols-radiation feedback" and "decrease in turbulent diffusion" to "the Explosive growth of PM2.5 mass" in Jing-Jin-Ji area, northern China. Numerical experiments are carried out for three runs, the first run absents "Aerosols-Radiation Feedback", the second run is with normal Aerosols-Radiation Feedback, and the third run is with reduced Turbulent Diffusion in addition. A one week haze event is modeled. Results of these runs, one by one, show improvement to reproduce the observed results.

**Response:**

We would like to heartily thank the reviewer for his serious review on our work and the valuable comments. We carefully considered comments of the reviewer and revised the paper accordingly, one by one of the following:

**My major concern and suggestion:**

**Comment 1)** This paper proposes a sensitive test on factors that influence the model result. But in the paper, results are directly presented, no middle results or any more supporting materials. Therefore, the conclusions are not convinced.

**Response:**

Thank the reviewer for this important comment. According to this comment, we revised the manuscript in following aspects:

Firstly, section 2.1 (line 90-132) is rewritten in the revised paper. The model description including dynamic, physical and some chemical processes is given in section 2.1. The parameterizing schemes and chemical mechanism used in this study and the related references are summarized in new Table 1 in the revised paper; An brief introduction of two-way coupling and the related references (line 113-119) and the calculation method of diffusion mixing in PBL scheme and the related references (line 124-132) are also added in the revised manuscript. Previous studies related with chemical process of the model (Gong and Zhang, 2007; Gong et al., 2012; Wang et al., 2010, 2015a,; Zhou et al., 2008, 2012, 2016) introduction in section 2.1 are added in the revised paper.

Secondly, Using hourly meteorology data from over 500 surface automatic observation stations of CMA, surface wind speed and temperature of Beijing, Xingtai and average in Jing-Jin-Ji by EXP1, EXP2 and EXP3 are evaluated. The modeled PBL wind speed and temperature are also studied (figure 3, the related discussion in line 230-260) and AERONET and CARSNET observed AOD (Table 4) and SSA (Table 5) are added to evaluated the model results (line 261-272). Study of downward shortwave fluxes due to AF and DTD (figure 5, 294-315) is also added to support the conclusions in the revised manuscript.

**Comment 2)** Reducing DC may lead the meteorological model running unrealistically. Details about the change of wind field etc. need to be displayed.

**Response:**

In our model, The DC is calculated in PBL scheme and it is passed into the chemical module (as DC_chem) to calculate the turbulence diffusion process of chemical tracers including gas and particles matter (PM). In our sensitive test, only DC_chem is reduced by 80% in the chemical module as a local variables but this change of DC was not changed in dynamic and other physics processed outside the CAUCE module. So, the turbulence diffusion process in PBL and wind in dynamic frame were not changed by the DTD sensitive experiment. The text line 162-178 is rewritten to explain this and the explanation of the three experiments. The explanation of this experiments set in table 2 is also corrected in the revised manuscript.

PBL meteorology background (figure 2) and wind and temperature changing (figure 3) are added to introduce and validate the meteorology condition of the haze episode in the revised manuscript, which also proved that the wind and temperature were not impacted by DTD.

**Comment 3)** Need description: synoptic background/weather condition for this haze event.

**Response:**

Figure 2 is added in the revised manuscript to show the geopotential height, wind and temperature at 500, 700, 850, 900, 950, 1000hPa to study the synoptic background and weather condition for this haze event. The added related text in line 206-223 in the revised manuscript.

**Comment 4)** Details of the model are needed, particularly the parts of lower atmosphere, levels, PBL scheme, surface model, radiation, aerosol absorption, etc.

**Response:**

The brief introduction of model dynamic, information of horizontal and vertical coordinates, physical package including PBL scheme, surface model, radiation etc. and chemical schemes, and the mechanism of aerosols direct and indirect mechanism are introduced in section 2.1 (line 89-160) and are also summarized in new Table 1 in the revised manuscript.

The introduction of two-way coupling including aerosols mixing method is also added in line 107-117 and the related references are also added in the revised manuscript.

Modeled aerosols optical depth (AOD) and single scattering albedo (SSA) representing the aerosol absorption are evaluated in the revised manuscript (table 4 and the related discussion)

**Comment 5)** PBL is mentioned as a crucial part in the paper, but no information about PBL is illustrated.

**Response:**

The introduction of DC calculation and PBL scheme and related references are added in line 118-126 in the revised manuscript.

The PBL meteorology background at 900, 950, 1000 hPa (figure 2) is also added in the revised manuscript. Figure 3 including PBL wind and temperature study are added in the revised manuscript. Figure 7 also showed the vertical structure of observation and modeled temperature, which included the information of PBL inversion.

**Other points:**

**Comment 1)** "Jing-Jin-Ji", not to be "JING-Jin-Ji" etc. different forms.

**Response:**

"JING-JIN-JI" and "JING-Jin_Ji" are all replaced by "Jing-Jin-Ji" in the revised manuscript.

**Comment 2)** Too many abbreviates, and their combination, hard to read the text; There are only 3 experiment runs, number them as Run 1□3, may be clearer.

**Response:**

"EXP1, EXP2 and EXP3" are used to replace the "EXP_bk, EXP_td_af, and EXP_td20_af" in the text, table and figures in the revised manuscript

**Comment 3)** Page 4, line 70-72:"One is that aerosols radiation feedback (AF) is not calculated online in the model run. AF can restrain turbulence by cooling surface and PBL while heating the atmosphere above it", Result of AF is mostly determined by absorbing aerosols, and by their vertical distribution.

**Response:**

This description is not accurate enough and it is revised as "AF may restrain turbulence by cooling surface and PBL while heating the atmosphere above it when aerosols with certain absorption characteristics concentrated in PBL" in the manuscript.

**Comment 4)** Page 4, line 77: "A Red-alert Heavy Haze occurred on 15 to 17 December", 15-23 Dec.

**Response:**

"15-17 Dec" is corrected as "15 to 23" in this line.

**Comment 5)** Page 4, Section 2.1, the model GRAPES_CUACE need to be introduced more detail, as well the setup of the simulations.

**Response:**

The detailed introduction of model GRAPES_CUACE is added in the section 2.1 including the related test and an added Table 1 including model dynamic frame and physical package in the revised manuscript; Line 148-159 in section 2.2 is rewritten to introduction the emission data in the revised manuscript and table 3 is added to list all VOCs emission used.

Section 2.4 Experiments Design (line 179-197) and table 3 are rewritten to introduce the setup of the simulations.

**Comment 6)** Page 5, Section 2.2, just lists the air pollutants, not relevant information crucial to this paper is given.

**Response:**

Line 148-159 in section 2.2 is rewritten to introduction the emission data in the revised manuscript and table 3 is added to list all VOCs emission used.

**Comment 7)** Page 5-6, Section 2.4, too simple in description. Table1, repeated, but still too simple.

**Response:**

Section 2.4 (line 179-197) and table 2 are rewritten to display the setup of the simulations in the revised manuscript.

**Comment 8)** Page6, line131:"which is named as the explosive growth (EG)", this is the first time mentions "explosive growth". Nothing is known what is the cause of EG: chemistry, transport, or accumulation of air pollutant?

**Response:**

From 00UTC on 17 to 00UTC 20 21 December, PM2.5 increased sharply and most of the study area reached the PM2.5 peaks of 400-600 ug/m3 rapidly during this period, which is named as the explosive growth (EG) stage (EGS) of PM2.5.

The cause of EG involves in several aspects such as meteorology, aerosols radiation feedback, chemistry, and transport etc. In this work, diffusion process of meteorology impacts and aerosols feedbacks were mainly discussed and regarded to contribute greatly to the PM2.5 EG. This is the main aim in section 3. The paragraph in line199-205 in section 3 is revised to explain this.

**Comment 9)** Page 6, Section 3.1, only PM2.5 is investigated. What about its source: primary or secondary? What about other pollutants?    And their effect on PM2.5 concentration?

**Response:**

Yes, there are many elements affecting PM2.5, such as emission, primary or secondary, gases and so on, but our study title is "The Contributions to the Explosive Growth of PM2.5 Mass…….". If we focus on the reason for the explosive growth of PM2.5, the atmosphere stable condition (turbulence diffusion) and the key elements what may result in distinct changes of it (AF) are the most important because the effects of primary or secondary aerosols and gas on PM2.5 concentration does not changes so greatly from clear day to PM2.5 EG stage during severe episode.

**Comment 10)** Page 7, Section 3.2, directly presents result of temperature profile, no logic description about the relation of AF and inversion strengthening. No qualitative and quantitative assessment on question if the result is right or correct.

**Response:**

Figure 6 in the revised manuscript is the vertical profiles of temperature changing due to aerosols feedback and it offered the qualitative and quantitative cause of the results of temperature inversion changing in Figure 7, line 323-337 is the explanation how the radiative cooling/heating rates due to aerosols resulted in the temperature inversion in figure 7 and offered quantitative temperature changes during CS and EG stage. Figure 7 displayed the observational and modeled temperature profiles and showed their obvious corrections by AF comparing with observation.

Anyway, we guess the reviewer want to know how the vertical profiles of temperature changing due to aerosols (figure 6) is calculated, so the detailed description of model introduction in section 2.1 is added to explain how the DT/dt_aero is calculated and impacts on model thermodynamics and then dynamic and physics.

**Comment 11)** Page 8, Section 3.3, the text is very difficult to read through since too many abbreviates.

**Response:**

The abbreviates "EGS,DC_bk, DC_td_af, DC_td20_af, PM2.5_bk, PM2.5_td_af, PM2.5_td20_af" are deleted and only the abbreviates "EXP1, EXP2, and EXP3" are remained in the revised manuscript.

**Comment 12)** Page 9, line 220-221: " significant decrease in turbulent diffusion on PM2.5 during EGS and DC_td_af was as low as 14m2/s on 20 December, which decreased about 50% comparing with DC_bk.", this sentence need to clarify. And "DC was 14m2/s", in where? What level? What time? Day or night?

**Response:**

This paragraph is corrected as "PBL DC at noon of EXP2 was as low as 14m2/s on 20 December, which decreased about 50% comparing with that of EXP1. PBL DC at noon of EXP2 on haze day was only about 20% of that on clear day. The PBL DC at noon……"

**Comment 13)** Page 10, line 245: "...we name it as 'turbulent intermittent'", What do you mean the 'turbulent intermittent'? Does 'turbulent intermittent' really mean lower diffusion coefficient or mixing rate?

**Response:**

When the turbulence diffusion processes is extreme weak and near zero turbulence, it is name "turbulent intermittent", in this study, when DC value is less than 4 to 6 m2/s, we consider it is near zero the turbulence diffusion named it as "turbulent intermittent".

A brief explanation is added in this line in the revised manuscript.

**Comment 14)** Page 10, line 253-254: "for the deficient description of extreme weak turbulent diffusion by PBL scheme in atmospheric models, are studied by analysing the changes of...", nothing about the PBL scheme is presented in this paper.

**Response:**

The introduction of DC calculation and PBL scheme and related references are added in line 124-132 in the revised manuscript.

**Comment 15)** in Table 1, "retaining 20% (reducing 80%) of normal turbulent diffusion", How to do this? Reducing the value at all the model domain?

**Response:**

The 80% reduction in turbulent diffusion coefficient (DC) is implemented in the chemical tracers (gas and particles) in the chemical module CUCAE. DC outside the CAUCE is not changed in the other parts of the model. Yes, The 80% reduction is applied to all simulated domain, but JING-JIN-JI region is mainly discussed in this study.

The solar radiation is the major cause of turbulence diffusion and PBLH diurnal changing during daytime. The observation study showed that the direct solar radiation on severe haze days is reduced 89% comparing with clear day in Beijing during the same period with this study (the following figure if from the result by Zhong, J.T., et al., 2018). The 80% reduction of turbulence diffusion is mainly according to this study. This reason is also added in section 2.4, Line 180-183; The changes of downward solar radiation fluxes and by AF+DTD is added in figure 5 (line 294-315) in the revised manuscript, which also support the supposing of 80% reduction of DC.

**Comment 16)** in Figure 5, the DC, at what position? What level/height?

**Response:**

Figure 5 in the initial version is figure 8 in the revised manuscript. and the DC is at 950 hPa, which is added in the following figure caption.

Fig.8 Hourly changing of PM2.5_OBS, PM2.5_EXP1, PM2.5_EXP2, and PM2.5_EXP3 (μg/m3), together with the turbulent diffusion coefficient at 950 hPa of the three experiments (DC_EXP1, DC_EXP2, DC_EXP3) from 15 to 22 December, 2016 in Beijing (a) and Xingtai (b)

**Anonymous Referee #2**

This paper investigated the impact of aerosol radiation feedback and decreased turbulent diffusion on PM2.5 during a heavy polluted episode in China. The objectives of this research might be interesting and potentially important; however, I have a number of concerns with the manuscript.

**Response:**

We would like to heartily thank the reviewer for his serious review and so detailed comments on our work. We carefully considered comments of the reviewer and tried our best to revise the paper accordingly, one by one of the following:

**General comments:**

**Comment 1:**

First, the lack of description about the GRPAES_CUACE model is troubling. What are the basic physical parameterizing schemes and chemical mechanism used in this study? How the model treat those crucial processes, such as SOA formation, two-way coupling, BC mixing states, aging processes. More important, how the model calculate the diffusion mixing? Any deficiency that can explain the supposed underestimation in diffusion coefficient, beside the lack of the aerosol radiative effect?

**Response:**

Thanks for this valuable comment. The section 2.1 (line 87-125) is rewritten in the revised paper according to this comment. The model description including dynamic, physical and some chemical processes is given in section 2.1. The parameterizing schemes and chemical mechanism used in this study and the related references are summarized in new Table 1 in the revised paper.

A brief introduction of two-way coupling and the related references (line 113-119) and the calculation method of diffusion mixing in PBL scheme and the related references (line 124-132) are also added in the revised manuscript.

Chemical processes involving such as SOA formation, BC mixing states, aging processes are very important to PM2.5 concentration, considering this content had been introduced and evaluated in previous studies they are not our major focus in this study (Gong and Zhang, 2007; Gong et al., 2012; Wang et al., 2010, 2015a,; Zhou et al., 2008, 2012, 2016). We add a brief introduction in section 2.1 to explain this and the offered the related references are added in the revised paper.

**Comment 2:**

Second, I suggest the authors to provide additional validation of the model performance. How was the model performance in simulating the meteorological variables, PM chemical components and precursors? Does the underestimation apply to all PM components? It is also very important to exam that how the change in diffusion influence on the model performance in simulating species including both PM chemical components and precursor, since the mixing process is critical in determining the concentrations of all species.

**Response:**

Yes, validation of the model performance is very important. The meteorology parameters close related with diffusion turbulence, such as surface and PBL wind speed and temperature (figure 3, the related text in line 225-260) and downward short wave fluxes (figure 5, the related text in 294-315) are added to provide the model performance and additional study in the revised manuscript. The three sensitive experiments are applied to all PM components.

Yes, mixing process is also critical in determining the concentrations of all aerosols species and precursor, but the discussion on PM chemical components and precursors are complex and will take up a great deal of space in the manuscript, considering the previous studies of the chemical processes by CUACE model (Gong and Zhang, 2007; Gong et al., 2012; Wang et al., 2010, 2015a,; Zhou et al., 2008, 2012, 2016) the focus of this study, observational aerosol optical depth (AOD) and single scattering albedo (SSA) in AEROSNET and CARSNET are closely related with chemical components (absorbing and scattering features) and direct impact on aerosols radiative feedback directly, so the two are added to evaluate the model performance in the revised manuscript (added table 4 and table 5, the related text in 261-272).

We are grateful for this valuable comment and will try our best to collect more observational data to focus on how the change in diffusion influence on the model performance in simulating species including both PM chemical components and precursor, since the mixing process is critical in determining the concentrations of all species in the following study.

**Comment 3:**

Third, the description about scenario design need be elaborated. In EXP_td_af, how the dynamic field is updated by the aerosol feedback, and is there any nudging processed? In EXP_td20_af, how was the 80% reduction in turbulent diffusion implemented in the model. Did the change apply to all simulated domains? Is there any evidence or references which can support such modification? Based on the results (overestimation is found for clean days and areas outside JJJ), I don't think the DTD is applicable for all grid cells and days and can explain the underestimation of PM2.5.

**Response:**

The mechanism of aerosols feedback on the dynamic is added in table 1 and the introduction of aerosol feedback and related references are added on line 113-119. There isn't nesting domain in the experiments.

The 80% reduction in turbulent diffusion coefficient (DC) is implemented in the chemical tracers (gas and particles) in the chemical module CUCAE. DC outside the CAUCE is not changed in the other parts of the model. Yes, The 80% reduction is applied to all simulated domain, but JING-JIN-JI region is mainly discussed in this study.

The solar radiation is the major cause of turbulence diffusion and PBLH diurnal changing during daytime. The observation study showed that the direct solar radiation on severe haze days is reduced 89% comparing with clear day in Beijing during the same period with this study (the following figure if from the result by Zhong, J.T., et al., 2018). The 80% reduction of turbulence diffusion is mainly according to this study. This reason is also added in section 2.4, Line 162-178; The wind speed changing (also an indicator of turbulence diffusion) from clear to haze days is added (figure 3 in the revised manuscript) in the revised paper, which also support the supposing of 80% reduction of DC.

[Figure]

Fig. 4. Daily radiant exposure of all selected clean days before 9 HPEs with CSs and all selected polluted days during the CSs of the HPEs. (a). Daily direct radiant exposure; (b). Daily diffuse radiant exposure; (b). Daily global radiant exposure

Yes, we agree that 80% reduction of DC is not simple applicable for all grid cells and days and can accurately explain all the underestimation of PM2.5 and out study did show DTD experiment is meaningful on PM2.5 EG in the Jing-Jin-Ji region. Anyway, we know even in Jing-Jin-Ji region, this study is only a sensitive experiment to explain the possible huge deficiency in the description of the extreme weak turbulence of the PBL scheme and 80% reduction may be not an accurate value in every grid point. It is very difficult at present to offer the direct proof of the truth turbulence diffusion condition leading to severe haze episode due to the lack of vertical PBL observations during daytime in this region. We added a short paragraph in the end in the conclusion section to explain the limitations of this study.

**Specific comments**

**Title**: need provide some description about "Red-alert" in introduction section

**Response:**

   The description about "Red-alert" is added in introduction section (Line 83-84) in the revised manuscript.

**Line 83:** "GRAPES_CUACE", provide the full name and some references about the model.

**Response:**

   An introduction of GRAPES_CUACE is rewritten and the related references are added (line 89-132) in the revised manuscript.

**Line 89:** How to get the boundary conditions?

**Response:**

   No boundary conditions or related text is discussed in this line, so we don't know what the meaning of this comment is.

**Line92:** "The model horizontal resolution is adopted as 0.15*0.15". Is it high enough to capture the strong inversion during the episode? What about the vertical resolution?

**Response:**

   The horizontal is optional in our model. Considering the resolution of emission inventory in China mainland obtained at present, 0.15*0.15 horizontal resolution is adopted in this study. If the model horizontal resolution is much higher than the resolution of emission data, model produces certain misleading results according to our experience. There are 33 vertical layers from surface to about 30 kilometers of the model top. Some introduction is added in line 96-97 in the revised manuscript. Our previous studies (Wang et al., 2015a; 2015b) showed that 0.15*0.15 horizontal resolution and the vertical layers used in this study had not much impact on the capturing of the strong temperature inversion.

**Line 100**: I would suggest the authors to elaborate the section 2.2. Is the emission data open to the public? What's the accuracy of the data? How does it compare to the others inventories, such as MEIC, EDGAR, etc?    How was the spatial / temporal allocation processed?

**Response**

   Yes, we couldn't give the complete and accurate description of the emission used in this study. The introduction of emission data including spatial and temporal information is rewritten in section 2.2 in the revised manuscript.

   In fact, we have long-term cooperation with MEIC team and may obtain the latest emission data from them. However, the emission condition in Jing-Jin-Ji region in China changed so rapidly, and our model is an operational haze forecast model in Chinese Meteorology Administration and we often find the MEIC emission data is time-lag for the real time forecasting, we had to do some corrections to MEIC emission data according to the latest emission reduction information in this region before using it.

   The emission data used in this study may be opened to the editor and reviewer, even to the public if this is required. We didn't use EDGAR emission data in our model also considering the rapid changes of emission in this region.

**Line 101:** "human life", is it "domestic"?

**Response:**

Yes, "human life" is replaced by "domestic" in this line.

**Line 105-106**: need provide full names for the VOC species

**Response:**

VOCs species listed \and the full names are also given in table 2.

**Line 121:** "a further 80% decrease in turbulent diffusion (DTD) of chemical tracers based on EXP_td_af representing a compensation for the insufficient description of extremely weak turbulent diffusion by PBL scheme in atmospheric chemical model". how the 80% decreased DTD was determined? Was the overestimation of vertical mixing is due to the coarse resolution, or underestimation of aerosol feedback?

**Response:**

The 80% reduction of turbulence diffusion is according to the reference by Zhong, J.T., et al., 2018 and the wind speed changing from clear to haze day (added figure 3 in the revised manuscript). In his study, the observation of direct downward short wave fluxes decrease about 89% in Beijing at the same period (This is the base of 80% DTD in section 2.4, the related explanation is added in section 2.4 in the revised paper). Even though, we know that 80% DTD is only a sensitive test and not a definite value in every grid point.

Even if the he coarse resolution do has some impacts on the vertical mixing, the impacts could not be so greatly only during the EG stage of PM2.5. We had been used a model 0.1*0.1 horizontal resolution and the results is basically same with the original. Aerosol feedback is one important reason, but not the all according to the results of the three experiments in this study.

**Line 134:** in section 3.1, what about PM chemical component? The mixing basically can revolve the total PM mass. However, if the chemical profile doesn't agree well the observation, it still cannot solve the issue.

**Response:**

Yes, PM chemical component is important, we can't find proper observational date of PM chemical components to compare with model outputs, considering observational aerosol optical depth (AOD) and single scattering albedo (SSA) are the important parameters related with chemical components and particle sizes (absorbing and scattering features) and its impacts aerosols on aerosols radiative feedback, AOD and SSA in AEROSNET and CARSNET stations are added to evaluate the model performance (added table 4 and table 5 and text in line 261-272 in the revised manuscript).

**Line 155:** "Some studies offline and online", is it "some offline/online modeling studies"?

**Response:**

Yes,this is revised in the manuscript.

**Line 157**: "AF of composite aerosols from black carbon, organic carbon, sulfate, nitrate, dust, ammonium, and sea salt aerosols had been online coupled into the in GRAPES_CAUCE model." how does the model treat mixing states and aging process? How is the model performance in simulating the PM components and AF?

**Response:**

The mixing method of black carbon, organic carbon, sulfate, nitrate, dust, ammonium, and sea salt aerosols was mainly introduced in previous study (Wang et al., 2015a). A brief introduction and the related references are also added in the 113-119 in the revised manuscript.

Observational aerosol optical depth (AOD) and single scattering albedo (SSA) are the important parameters close related with chemical components (absorbing and scattering features) and they are also define the AF effects directly, AOD and SSA in AEROSNET and CARSNET stations are added to evaluate the model performance (added table 4 and table 5 and text in line 261-272 in the revised manuscript).

**Line 173:** "the temperature inversion layer pre-existed during the haze event", it is not easy to see the temperature inversion in the plots.

**Response:**

It is easy to see in figure 7a and figure 7b, not in figure 6 in the revised manuscript. There is similar phrase in the discussion on figure 7a and figure 7b, so, this phrase is deleted in the revised manuscript.

**Line 182:** "Figure 4b shows that the observed temperature inversions were obvious stronger and the inversion depth thicker on 18 to 19 (during EGS of PM2.5) than those on 15 to 16 Dec (CS of PM2.5"    But the PBL height seems opposite, lower on 18 to 19 but higher on 15 to 16 Dec.

**Response:**

No PBL height was displayed in this study. We are not sure where the reviewer drew the conclusion "But the PBL height seems opposite, lower on 18 to 19 but higher on 15 to 16 Dec"

According our previous studies (Wang et al., 2015a, 2015b), when the temperature inversion is stronger, the corresponding PBL height is lower and PM2.5 is higher.

**Line 191:** "The contributions to PM2.5 EG due to AF and DTD". Since AF also contributes to DTD, how to separate these two effects.

**Response:**

The contribution to PM2.5 due to AF means the PM2.5 changing due to aerosols feedback online (EXP2 in the revised paper), only including the diffusions reduction by AF, but not including 80% reduction of DC; The results of DTD (EXP3 in the revised paper) means the differences between EXP3-EXP2, it does not include the AF's contribution, but only the decrease of turbulence diffusion of coefficient of chemical tracers. In EXP3, The DTD is implemented in the chemical tracers (gas and particles) in the chemical module domain. DC outside the CAUCE is not changed in the model run.

**Line 207:** "Exp_bk under underestimated the PM2.5", "under" should be deleted

**Response:**

"under" is deleted in the text.

**Line 224:** "the overestimation of turbulent DC", is there any observation data to prove the overestimation of DC?

**Response:**

The solar radiation is the major cause of turbulence diffusion and PBLH diurnal changing during daytime. The observation study showed that the direct solar radiation on severe haze days is reduced 89% comparing with clear day in Beijing during the same period with this study (the following figure if from the result by Zhong, J.T., et al., 2018). The 80% reduction of turbulence diffusion is mainly according to this study. This reason is also added in section 2.4, Line 180-183;

The changes of downward solar radiation fluxes and by AF+DTD is added in figure 5 (line 294-315) in the revised manuscript, which also support the supposing of 80% reduction of DC.

**Figure 2:** The PM2.5 in area outside JJJ seems all overestimated. The td_af cases make it even worse. Seems like it is not proper to apply the 80% DTD to all grid cells.

**Response:**

PM2.5_obs is the station observation data and the each color dot represents the value in the station, the white color stands for lack of observation data not the lower PM2.5 value < 35ug/m3, which is not completely same with the modeled PM2.5 on grid points with high resolution. Excluding this reason, PM2.5 by EXP3 (td20_af in initial manuscript) is still the best in general, then EXP2 (td_af), and EXP1 is the worst in Jing-Jin-Ji comparing with observation PM2.5. Outside JJJ, td_af cases make it worse in the area with low PM2.5, make it better in the area with higher PM2.5. Anyway, this study mainly focuses on Jing-Jin-Ji region.

Certainly, we agree 80% DTD may be not accurate to all grid cells even in Jinh-Jin-Ji region. Our study area is Jing-Jin-Ji and even in this area the 80% DTD can't represents the exact condition of turbulence diffusion in all grid cells. Our study is sensitive experiment and we hope the underestimation of high PM2.5 due to the distinct deficiency of PBL scheme in the description of the extreme weak turbulence diffusion in Jing-Jin-Ji in east China may cause attention by this sensitive experiment. The final solution for this underestimation depends on the improving of PBL algorithm base on more detailed observation of PBL meteorology scales, not the simple decreasing of DC.

A paragraph is added in the last in section 4 to explain all above limitations and the other possible reasons leading to the underestimation in this study.

**Figure 3:** please clarify that the data is regional average in JJJ.

**Response:**

This is revised in the caption this figure (figure 6 in the revised version).

**Figure 4:** what about the days when PM reach peak for Dec 20-22 in Beijing.

**Response:**

The inversion and the impacts on it due to AF are similar in 20-22 with that in EG stage. The explanation about this is added in the text after this figure.

**Figure 5:** PM2.5_td_af seems more reasonable than PM2.5_td20_af, in consideration of the possible missing heterogeneous chemistry. What's the reason for the underestimation of the peak on Dec 21, even though the DC is already very low.

**Response:**

CUACE model includes a simple scheme of heterogeneous chemistry of SO2 and the related explanation is added in section 2.1 in the revised model.

Yes, "heterogeneous chemistry" is a very important influencing factor to PM2.5 concentrations, but there are also many uncertainties of this influence due to a series of complex chemical processes and species. At present, it is very difficult to offer a quantitative estimation of the impacts of heterogeneous chemistry on PM2.5 either in observation or in model.

There are several causes impacting local PM2.5 concentration involving in emission, meteorology, atmospheric chemical processes in including gas-particles and "heterogeneous chemistry" and etc. Some studies emphasize the impacts of meteorology condition including the feedback from AF. Some studies stressed the impacts of heterogeneous chemistry on PM2.5. It's a controversial issue. This study mainly focuses on meteorology impacts from turbulence diffusion and aerosols feedback. Anyway, this is a limitation of this study and it is explained in the last paragraph in section 4 in the revised paper.

The PM2.5 and DC condition on December 21 mainly related with changing of meteorology condition such as the inversions and wind fields, a sort explanation is added in this paragraph to explain it.

**Figure 6**: the figure is misleading. Since the reduced error in td20_af is because that the overestimation on Dec 18 compensates the underestimation on Dec 21 in Beijing.

**Response:**

The result of this figure is calculated by the model result from 00 UTC 17 to 00 UTC on 21 December, the data in 21 December is not included in the calculation. The description of CS and EG is not accurate and it is corrected in section 3.1 in the revised paper.

**The Contributions to the Explosive Growth of PM$_{2.5}$ Mass due to Aerosols-Radiation Feedback and Further Decrease in Turbulent Diffusion during a Red-alert Heavy Haze in Jing-Jin-Ji in China**

Hong Wang[1,2 *], Yao Peng[1,2], Xiaoye Zhang[1,3*], ,Hongli Liu[1], Meng Zhang[4], Huizheng Che[1], Yanli, Cheng[1]

State Key Laboratory of Severe Weather (LASW), Chinese Academy of Meteorological Sciences (CAMS), CMA, Beijing 100081, China

Collaborative Innovation Center on Forecast and Evaluation of Meteorological Disasters, Nanjing University of Information Science & Technology, Nanjing 210044, China

Center for Excellence in Regional Atmospheric Environment, Institute of Urban Environment, Chinese Academy of Sciences (CAS), Xiamen 361021, China

Beijing Meteorological Bureau, Beijing 100089, China

Correspondence to: Hong Wang (wangh@cma.gov.cn),Xiaoye Zhang (xiaoye@ cma.gov.cn)

**Abstract.**The explosive growth (EG) of PM$_{2.5}$ mass usually resulted in PM$_{2.5}$ extreme levels and severe haze pollution in east China and they were generally underestimated by current atmospheric chemical models. Based on the atmospheric chemical model GRPAES_CUACE, three sensitive experiments of background (EXP1),  aerosols feedback online (EXP2), and decrease 80% in turbulent diffusion coefficient (DTD)  of chemical tracers  based on EXP2 (EXP3) are designed to study the contributions to the EG of PM$_{2.5}$ due to aerosols-radiation feedback (AF) and  focusing on a red-alert heavy haze  in Jing-Jin-Ji region of China. The study results showed that turbulent diffusion coefficient (DC) calculated by EXP1 is about 60-70m$^2$/s on clear day and 30-35m$^2$/s on haze day. This difference of DC was not enough to discriminate the unstable atmosphere on clear day and extreme stable atmosphere during EG stage of PM$_{2.5}$, and the inversion calculated by EXP1 was obviously weaker than the actual inversion from  sounding observation on haze day. This led to 40-51% underestimation of PM$_{2.5}$  by EXP1; AF reduced about 43-57%  DC during EG stage of PM$_{2.5}$, which strengthened the local inversion obviously and the local inversion by EXP2 was much closer to the sounding observation than that by

EXP1. This resulted in 20-25% reduction of model negative errors of PM$_{2.5}$ and it was as low as

-16 to -11% in EXP2. However, the inversion by EXP2 was still weaker than the actual observation and AF could not solve all the problems of PM$_{2.5}$ underestimation. Based on EXP2,

80% DTD of chemical tracers in EXP3 resulted in a near-zero turbulent diffusion named as "turbulent intermittent " atmosphere state, which EXP3  resulted in further 14-20%

reduction of PM$_{2.5}$ underestimation and the negative PM$_{2.5}$ errors of was reduced to -11 to 2%

The combined effects of AF and DTD solved over 79% underestimation of PM$_{2.5}$ EG in this  study. The results show that the online calculation of aerosol-radiation feedback is essential for the prediction of PM$_{2.5}$ EG and peaks during severe haze in Jing-Jin-Ji region.  Besides this, an improvement in the arithmetic of PBL scheme focusing on extreme stable atmosphere stratification is also indispensable for reasonable description of local "turbulent intermittent" and more accurate prediction of PM$_{2.5}$ EG  during the severe haze in Jing-Jin-Ji in China.

[revised manuscript text omitted]

**2 Model, Data and Methodology**

**2.1 GRAPES_CUACE Model**

Focusing on dust and haze pollutions in China and East Asia, the Chinese Unified Atmospheric Chemistry Environment (CUACE) (Gong and Zhang, 2008) was online integrated into mesoscale version of Global/Regional Assimilation and PrEdiction System (GRAPES_meso) developed by the Chinese Academy of Meteorological Sciences (Chen et. al., 2008; Zhang and Shen, 2008) to build an online chemical weather forecasting modelThe double way atmospheric chemical model GRAPES_CUACE (Wang et al., 2009, 2010; 2015a; Zhou et al., 2012)was established focusing on simulation and prediction of dust and haze pollutions in China and East Asia. The main components of GRAPES_CAUCE include: model dynamic core; modularized physics package (Xu et al., 2008), atmospheric chemistry module CUCAE with online coupling of aerosols direct and indirect feedback and emission inventory. The dynamic frame of GRAPES_CUACE is semi-implicit semi-Lagran full compressible nonhydrostatical (Yang et al., 2007, 2008; Chen et al., 2008). A height-based-terrain following coordinate was used and there are 33 vertical layers form surface to 30 kilometers. The longitude-latitude grid is adopted in the spatial discretization of and the horizontal resolution is optional. The physical packages is ptional (Xu et al., 2008) and table1 lists the specific physics and chemistry schemes used in this study. Gas-phase chemistry of RAD II (Stockwell et al., 1990) with 63 gaseous species through 21 photo-chemical reactions and 121 gas phase reactions is used in this study. The aerosols includes sea salts (SS), sand/dust (SD), black carbon (BC), organic carbon (OC), sulfates (SF), nitrates (NI) and ammonium salts (AM) and aerosols processes involving in hygroscopic growth, coagulation, nucleation, condensation, dry and wet depositions, scavenging, aerosol activations and etc. The formation of sulfate aerosols and second organic aerosols (SOA) from gases, nitrates and ammonium formed through gaseous oxidation, and ISORROPIA (Fountoukis et al., 2007) calculating the thermodynamic equilibrium between nitrates and ammonium and their gas precursors are considered in CAUCE, which had been evaluated and introduced in previous studies. (Gong and Zhang et al., 2008; Zhou et al., 2008, 2012).

Based on the modeled aerosols concentration, vertical profiles of temperature changing including aerosols direct impacts (DT/dt due to aerosols) is calculated by radiation model and online feedback to the model dynamic core in each grid point in every time step, which reforms model temperature field, dynamic process, regional circulation and meteorology condition, finally impacts aerosols concentration in turn. The external mixing of aerosols species of SS, SD, BC, OC, SF, NI, and AM and particle size bins is used in the calculation of aerosols radiation feedback, which was introduced and evaluated in detail in previous studies (Wang et al., 2009, 2010, 2015a, 2015b). With this double way GRAPES_CUACE model, Trans-city and regional transportation of PM2.5, aerosols aerosols-radiation-PBL-meteorology interactions, and aerosols-cloud-precipitation interactions etc., and regional pollution and transportation of $PM_{2.5}$ etc. had been widely successfully simulated and studied by using it (Wang et al., 2009, 2010, 2015a, 2015b; Zhou et al., 2012, 2016; Jiang et al., 2015; Zhang et al., 2018). GRAPES_CUACE is also used in this study.

The turbulent diffusion coefficient (DC) is calculated by YonSei University (YSU) PBL scheme (Hong et al., 2006), which is a revised vertical diffusion package based on nonlocal boundary layer vertical diffusion scheme in a Medium-Range Forecast model (MRF) (Hong et al., 1996). The major ingredient of the revision is the inclusion of an explicit treatment of entrainment processes at the top of the PBL comparing with MRF PBL scheme. The specific calculation method of DC was show in Hong's studies. This algorithm of DC was has been widely selected as a standard option for the Medium Rang Forecast (MRF) Model (Caplan et al. 1997; Farfán and Zehnder, 2001; Basu et al., 2002; Bright and Mullen, 2002; Mass et al., 2002) and Weather Research and Forecast (WRF) model (Hong et al., 2006) in National Centers for Environmental Predictions (NCEP) since its establishment.

The model model horizontal resolution is adopted as 0.15°×0.15° to match the resolution of emission source data used in this study. Considering the impacts of interregional transport of gas and particle pollutants, in the main polluted areas in eastern China, the model domain includes the whole east China (100-140°E, 20-60°N) (figure 1a) was set as the model domain, but our study discussion mainly focuses on the most polluted area Jing-Jin-Ji region (the red box in figure 1a). ) and Figure figure 1b shows the detailed features of geographical location and topography of JING Jing Jin Jithis region. The black dots in Figure1a are the locations of $PM_{2.5}$ observation stations. The model horizontal resolution is adopted as 0.15°×0.15° to match the resolution of emission source data used in this study. There are two balloon sounding stations, Xingtai and Beijing (yellow stars in Figure figure 1b) in our study area. Xingtai, located in southern Hebei province, the eastern foot of Taihang Mountains and it is influenced by the sinking airflow from Taihang Mountains in winter, is the most polluted city and the $PM_{2.5}$ concentrations usually ranked the first in China in recently years. The topography of Xingtai and the serious haze pollution closely related to it are is the typical representative of the southern plain of Jing-Jin-Ji. Beijing lies in the transitional zone from Yan Mountain to its southern plain, next to Tianjin and surrounded by Hebei, representing the polluted areas in the central part of Jing-Jin-Ji.

**2.2 Emission Inventory**

Based on MEIC emission inventory in 2012 (He et al., 2012), the changes of 5 kinds of emission sources of industrial, human lifedomestic, agricultural, natural and traffic are obtained by from the data statistics data of China national industry factories, energy consumption, road net and motor vehicles, population information, land use, vegetation cover and etc. in 2015 and 2016 are updated to 2015 to 2016

in east China.

5 reactive gases, i.e. $SO_2$, NO, $NO_2$, CO, $NH_3$, 20 VOCs, i.e. ALD, $CH_4$, CSL, ETH, $HC_3$, $HC_5$, $HC_8$, HCHO, ISOP, KET, NR, $OL_2$, OLE, OLI, OLT, $ORA_2$, PAR,

TERPB, TOL, XYL and ( VOCs species listed in table 2 ) and 5 aerosols species, i.e. black carbon, organic carbon, sulfate, nitrate and fugitive dust are obtained by above emission data according the input requirement of CUACE model. The horizontal grid resolution is 0.15°×0.15° and there is one emission data set for each month with hourly interval.

**2.3 Data Used**

Hourly  observation $PM_{2.5}$  concentration data for more than 1440 surface observational stations (blue dots in figure 1) from China National Environmental Monitoring Centre (CNEMC)

(http://www.cnemc.cn) from 15 to 23 December 2016 were used to evaluate the model results; The hourly observation meteorology data including wind speed, and temperature from 500 surface automatic observation stations in China Meteorology Administration (CMA) in Jing-Jin-Ji region (red triangle in figure 1b) were used to model validation.

The meteorological balloon sounding data at 00UTC (early morning) and 12UTC

(and dusk in local time) in  Beijing and Xingtai (yellow star in figure 1b) from (CMA during the same period were also used compare with the modeled results; There are one AERONET station (Holben et al., 1998) Xianghe, and two CARSNET stations (Che et al., 2009; 2014; 2015) Beijing and Shijiang in Jing-Jin-Ji region (black crosses in figure 1b). Observed aerosols optical depth (AOD) and single scattering albedo (SSA) date from the three stations at the same time period were also used to model evaluation; NCEP 0.25×0.25° global analysis grids data (https://rda.ucar.edu/datasets/ds083.3) were used as the model initial and every 6-hour lateral boundary meteorology input fields. The initial values of chemical tracers were obtained according to the five-year mean climatic values. The results of the first 120 hours of model start are split out to eliminate the effects of chemical initial fields.

**2.4 Experiments Design**

Both dynamic process of regional atmosphere and solar radiation  have  important impacts on turbulence diffusion and PBL processes. When severe haze occurred, it was observed that the surface daily direct radiant exposure was observed reduction 89%  comparing with that on clean days (Zhong et al., 2018), suggesting the possible huge difference of turbulence diffusion between severe haze and clean days. It is difficult to distinguish the two reasons leading to the extreme weak turbulence diffusion in the truth atmosphere because of the complicated relationship between atmosphere dynamic and solar radiation. However, some meaningful research could be expected by sensitive experiments using atmosphere chemical model. Three sensitive experiments of EXP1EXP1, EXP2, and EXP3EXP3  are designed to discuss the  contributions to $PM_{2.5}$ EG the extreme weak turbulence and corresponding $PM_{2.5}$ EG due to AF and the insufficient description on the extremely weak turbulent diffusion by PBL scheme in atmospheric chemical model. EXP2  (the detailed escriptions of the three experiments listed in Table 3). All other model dynamic process, physical options and initial input data of meteorology and chemical tracers are same for the three experiments except for the differences shown in Table 3. In the sensitive test in EXP3,further decrease in turbulence diffusion coefficient (DTD ) based on EXP2 was only applied to the DC of chemical tracers in CUACE mode and DC in other physical packages and dynamic frame of GRAPES_MESO was same with that in EXP1 and EXP2.

**3 Results and Discussions**

This haze episode began on 15 December, 2016.  $PM_{2.5}$ began to gather and climb slowly at this time but it was below 150 ug/m$^3$ in most Jing-Jin-Ji region from 00UTC on 15 to 00 UTC on 17 December, and we name this period as the climbing stage (CS) of $PM_{2.5}$; From 00UTC on 17 to 00UTC  21 December, $PM_{2.5}$ increased rapidly , and reached the $PM_{2.5}$ peaks of 400-600 ug/m$^3$ in  most of the study area. This period is named as the explosive growth (EG) stage  of $PM_{2.5}$. This section mainly focuses on the contributions to the $PM_{2.5}$ EG due to AF and further DTD.

**3.1 The synoptic background of the haze episode**


The upper atmosphere circulation and surface synoptic system controlling Jing-Jin-Ji region remained relatively stablehas not changed much during the whole haze maintenance. Figure 2 displayes Geopotential height (GPH), temperature (Temp) and Wind fields at high (500hPa), middle (700hPa), low atmosphere (850hPa) and PBL levels (900, 950, 1000hPa) on 00 UTC, 19 December, 2016 as the typical representative to showing the weather background of this haze event. It is can be seen that GPH in the upper atmosphere (500hPa) showed zonal circulation in East Asia. There was a horizontal trough north to Jing-Jin-Ji (black box) in the upper and middle atmosphere (500 and 700 hPa) and Jing-Jin-Ji was controlled by the weakmoderate northwest or west air flow at the bottom of the trough. Temperature and wind fields at 500 and 700hPa both showed that cold air in the upper and middle atmosphere was weak. GPH in 850hPa showed that the subtropical high (SH in figure 2) in east sea was strong and Jing-Jin-Ji was in the pressure equalization field to the northwest periphery of the subtropical high and the wind was very weak in this level due to the block of the subtropical high. GPHs at 900, 950, 100hPa all showed that Jing-Jin-Ji located in the pressure equalization field between the northwest land high (LH in figure 2) and southeast subtropical high within the whole PBL and the land high was weaker than the subtropical high. This resulted in small pressure gradient, weak and thin wind fields and stable atmosphere situation within PBL in Jing-Jin-Ji region, which is very helpful to the maintenance of haze episode.

**3.1 2 The Comparison study of observation and model results**

Not only surface but also PBL meteorology are the key factors affecting the process haze episode and $PM_{2.5}$ level (Wang et al., 2014a, 2014b), but it is well known that surface and PBL meteorology factors are more difficult to be predicted or simulated by most numerical models than those at middle and high atmosphere, which is also the key point affecting the prediction performance of atmospheric chemical models (Hu et al., 2013a, 2013b; Li et al., 2016).

Using hourly meteorology data from surface automatic observation stations of CMA, surface wind speed and temperature of Beijing, Xingtai and average in Jing-Jin-Ji by EXP1, EXP2 and EXP3 are evaluated from 15 to 24 December, 2016 (figure 3, up). It can be seen that in Beijing, the modeled surface wind speed by the three model experiments was in good agreement with the observation regardless of the changing trend, maximum and the minimum values of wind speed. The observed and modeled wind speed was basically below 2 m/s from 17 to 21 December (EG stage of $PM_{2.5}$). Modeled wind speed in Xingtai was slightly worse than those in Beijing, but the changing trend of wind speed was basically consistent with those of observation and the wind speed was also below 2 m/s during the EG stage of $PM_{2.5}$. The modeled wind speed was higher than observation to a certain extent at the beginning and ending period in Xingtai. The changing trend of modeled average wind speed in Jing-Jin-Ji region showed reasonable agreement with that of observation and was the closet to the observation at the EG stage of $PM_{2.5}$. The regional wind speed by model was higher than observation in general. The comparison of wind speed of the three model experiments showed that the wind speed by EXP2 and EXP3 was basically same, but both smaller than EXP1 in various degree in Beijing, Xingtai, and average in Jing-Jin-Ji during EG stage, showing that AF decreased surface wind speed. The temperature changing trend by the three model experiments also consisted with that of the observation on the whole in Beijing, Xingtai and Jing-Jin-Ji. But it also can be seen that the modeled temperature was obvious higher than observation, especially during the EG stage. The temperature by EXP2 and EXP3 was basically same, but lower than that by EXP1, which is much closer to the observation, indicating that AF reduced the positive errors of surface temperature in Beijing, Xingtai, and average in Jing-Jin-Ji. However, it can be seen that the temperature by EXP2 and EXP3 was also higher than observation during the EG stage, suggesting that some other uncertainties in PBL scheme led to the temperature positive errors during EG stage besides AF, which deserves further study in detail. PBL mean wind of the three experiments in Beijing, Xingtai, and regional average in Jing-Jin-Ji were calculated and shown in figure 3 (down). Unfortunately, there are not observation data to evaluate them. Comparison of the PBL wind and temperature of the three model experiments showed that PBL mean wind was basically below 4m/s while the temperature is high at the EG stage in Beijing, Xing tai and Jing-Jin-Ji. Similar to the ground results, the PBL mean wind speed and temperature by EXP2 and EXP3 were basically same, but the wind speed by the two experiments was obviously lower than that by EXP1. This indicated that the reduction of wind speed by AF was more obvious in PBL than that in ground,while comparison of surface and PBL temperature of the three experiments showed that the cooling effect by AF is much stronger at surface than that in PBL.

Aerosols optical properties including AOD, SSA, and asymmetry factor (ASY) largely determines the aerosols direct radiation effects. The observed AOD (Table 4) and SSA (Table 5) in Shijiazhuang, Beijing and Xianghe are used to evaluate the modeled results from 15 to 22 December. Because the differences of the modeled AOD and SSA by the EXP1, EXP2 and EXP3 are small, the results of EXP1 are used here. It can be seen that the values of modeled AOD and SSA and their temporal changing trend from 15 to 22 December were basically consistent with the observation in Beijing, Shijiazhang and Xinghe, proving the model performance in the description of aerosols optical properties. Both observed and modeled SSA in Shijiazhuang, Beijing, and Xianghe (table 5) shows that SSA was obvious higher during the EG stage of $PM_{2.5}$ than that at the beginning or ending stage of haze on 15 to 16 and 22 December, illustrating that the scattering characteristics of composite aerosols increased obviously when high AOD and $PM_{2.5}$ occurred on severe haze days in Jing-Jin-Ji region. The accurate description in AOD and SSA, especially the SSA changing from clean to haze days, is the basic in the following discussion of aerosols effects on $PM_{2.5}$.

Figure  4 displays the averaged observed $PM_{2.5}$ ($PM_{2.5}$_OBS) and simulated $PM_{2.5}$ of  EXP1($PM_{2.5}$_EXP1), EXP2 ($PM_{2.5}$_EXP2) and EXP3($PM_{2.5}$_EXP3) experiments during EG stage. It can be seen from $PM_{2.5}$_OBS that the averaged $PM_{2.5}$ values were generally over 100μg/m$^3$ in east China and Jing-Jin-Ji covered the most polluted areas and $PM_{2.5}$ reached up to 300 to 400μg/m$^3$ in parts of Beijing, Tianjin, Middle-south Hebei province, western frontier region of Shandong province and north Henan province. The $PM_{2.5}$ center of 500-700μg/m$^3$ appeared in south Hebei and North Henan province and the $PM_{2.5}$ maximum of _700μg/m$^3$ was found in south Hebei. The comparison study of $PM_{2.5}$_EXP1 and $PM_{2.5}$_OBS shows that $PM_{2.5}$_EXP1 is obvious lower than $PM_{2.5}$_OBS on the whole. It is noteworthy that  EXP1 fail to simulate the $PM_{2.5}$ over 300μg/m$^3$. $PM_{2.5}$_OBS is about 200 to 300μg/m$^3$ over most Shandong province while the $PM_{2.5}$_bk is only 100 to 200μg/m$^3$ in this region. Compared with $PM_{2.5}$_EXP1, $PM_{2.5}$_EXP2 values are significantly improved by AF and they are much closer to the $PM_{2.5}$_OBS. High $PM_{2.5}$_OBS centers of 300 to 400, 400 to 500, and 500 to 600μg/m$^3$ are almost simulated by EXP2, indicating the important effects of AF on the model simulation of $PM_{2.5}$ high values. However, the areas of the simulated

PM$_{2.5}$ values of 300 to 400, 400 to 500, 500 to 600μg/m$^3$ are still smaller than that of the PM$_{2.5}$_OBS. EXP2 also fails to simulate the maximum PM$_{2.5}$ values over 600μg/m$^3$ observed in south Hebei province. PM$_{2.5}$ EXP3 just makes up for this shortage, comparing with PM$_{2.5}$ EXP1 and PM$_{2.5}$ EXP2, PM$_{2.5}$ EXP3 is undoubtedly the closest to PM$_{2.5}$_OBS both in PM$_{2.5}$ extreme and its influence area. This study result illustrates that both AF and DTD in atmospheric chemical models are required for the effective prediction of PM$_{2.5}$ EG during the severe haze in Jing-Jin-Ji in China.

**3.3 The downward solar radiation flux change by aerosols and DTD**

PM in the atmosphere will inevitably lead to the changes of surface and atmosphere solar radiation flux. When severe haze occurs, most PM is concentrated in the atmosphere near the surface and within PBL, solar radiative flux reaching the ground is reduced greatly, which is the direct trigger factor for the subsequent changes in thermodynamic, dynamics, and then atmospheric stratification. Any factor leading to the change of the atmosphere PM loading might result in change of the surface downward solar radiation flux (SDSRF). We calculated the percentage changes of SDSRF (W/m$^2$) between EXP2 and EXP1 ((SDSRF$_{EXP2}$-SDSRF$_{EXP1}$)/SDSRF$_{EXP1}$) and EXP3 and EXP1 ((SDSRF$_{EXP2}$-SDSRF$_{EXP1}$)/SDSRF$_{EXP1}$)) to study the impacts on SDSRF by aerosols and DTD. Figure 5 shows the mean percentage change of SDSRF (W/m$^2$) due to aerosol (a) and aerosol plus DTD (b) of EG stage. It can be seen that SDSRF was reduced more than 50% by aerosol in most study region, 60-65% in Jing, Jin, most of Ji, and Northern Shandong, even 65-70% in Jing, Jin, and part of Ji, indicating the important influence of aerosols on SDSRF. Comparison of figure 5b and 5a showed that this reduction of SDSRF by aerosol (figure 5a) in EXP2 was further strengthened by DTD of chemical tracers in EXP3 (figure 5b) in certain region because DTD made more PM$_{2.5}$ gather near surface (figure 3), transport less and this led to the increasing of total PM$_{2.5}$ loading. It also can be seen that the difference of figure 5a and figure 5b was not too much. This is because that the major impacts of DTD is to reform the vertical distribution of atmosphere loading of PM$_{2.5}$ and its impacts on total column of PM$_{2.5}$ is not so much. On the other hand, the reduction of SDSRF due to aerosols radiation was already very great, and the change of SDSRF due to the increased column PM$_{2.5}$ by DTD, would not be so great on a secondary basis. This value of the SDSRF reduction due to aerosols and

DTD is basically consistent with the 56-89% difference of observational radiant exposure between clear and haze day at the same period (Zhong et al., 2018).

**3.2 4 The aerosols' reform on local atmosphere temperature profiles**

 Offline and online studies  indicated the reforming of atmosphere temperature profile  by aerosols direct radiation (Wang et al., 2010, 2015b; Forkel et al., 2012; Gao et al., 2014, 2015; Wang et al., 2014; Gao et al.,  2017; Ding et al., 2016). In our previous works (Wang et al., 2015a, 2015b),  composite aerosols mixing black carbon, organic carbon, sulfate, nitrate, dust, ammonium, and sea salt aerosols had been online coupled into the in GRAPES_CAUCE model. On this basis, the changes of mean temperature profile of Jing-Jin-Ji region of daytime due to aerosols radiation were calculated from 15 to 20 December, 2016 in this work. It can be seen from Figure  6 that  aerosols cooled the atmosphere below 750 to 800 hPa while warmed the atmosphere above this height. Considering PBL height may be as low as several hundreds to one thousand meters when severe hazes occurs in Jing-Jin-Ji (Wang et al., 2015a, Zhong et al., 2017), it may be concluded that whole PBL and its near upper atmosphere was cooled by  aerosols to a different extent during the different stage of this haze. The aerosols' warming effects above 750-850hPa height were very weak and the temperature changes among different days were also small. However, the aerosols' cooling effects shows the most differences from surface to 975 hPa height on different day. The surface daytime cooling is about 2.2 K on 19, 1.5K on 18 and 20, 1K on 17, and 0.5-0.6 K on 15 to 16 December. This aerosols' cooling effect decreased rapidly with the height. The difference of cooling rates between surface and 850hPa is 1.8 K on 19, 1.3K on 18 and 20, 1K on 17, and 0.3-0.4 K on 15 and 16 December.  The  difference of cooling rates by aerosols between surface and upper PBL are much bigger during  EG stage  than  that during  CS.  This may result in the further intensification of the temperature inversion layer pre-existed during the haze event, which will be discussed in figure 7 in the following section.

The vertical sounding meteorology data in Beijing and Xingtai    can be used to prove if this change of the temperature profile by  aerosols is correct or not. Figure  7 shows the vertical temperature profiles of sounding observation and the modeled temperature profiles  by

EXP1 and EXP2 during CS (Figure 7a) and EG stage (Figure 7b) at the two stations. The temperature profiles (Figure 7a) shows that both modeled results by EXP1 and EXP2 partly simulated the observed temperature inversion in Beijing and Xingtai on 15 to 16. The very little difference between the temperature profiles  by EXP1 and EXP2 indicated that aerosols radiation had very little impacts on the temperature profiles and local inversion during the CS of $PM_{2.5}$. Nevertheless, Figure 7b shows that the observed temperature inversions were obvious stronger and  thicker on 18 to 19 (EG stage) than those on 15 to 16 (CS of $PM_{2.5}$) both in Xingtai and Beijing. The temperate profiles by EXP2 were much closer to the observation results than that by EXP1, and especially, the temperature inversions were much stronger and also closer to the observation than that by EXP1. This result proved that the effective correction of local inversions by aerosols during the EG stage of $PM_{2.5}$.

However, it also can be seen, that the inversions by EXP2, which included online AF, are still weaker than the truth observed inversion in the two stations. This  suggests that there must be other causes for the underestimation of the observed extreme strong inversion  by the model besides the online calculation of AF, which is worthy of studying.

**3.5 The contributions to $PM_{2.5}$ EG due to AF and DTD**

Turbulent diffusion process is the main way of gas and particles exchanging from ground to upper atmosphere and then removed by the high altitude transport, which is usually  achieved by turbulent diffusion process  in the chemical atmospheric models. Firstly, the inversion and weak turbulent diffusion, which generates from atmosphere dynamic process, leads to atmosphere stabilization and determines the occurrence of haze and its strength (Zheng et al., 2016). Once the haze occurs, the aerosols radiation may reinforce the inversion in turn when aerosols exceeds certain critical value and lead to more $PM_{2.5}$ gathering near the ground . The relative importance of the two aspects on $PM_{2.5}$ EG may vary with the $PM_{2.5}$ values and meteorology conditions, but they are irreplaceable for the reasonable prediction and simulation of $PM_{2.5}$ EG and peaks by atmospheric models.

Figure 8 displays the hourly changing of observed $PM_{2.5}$ ($PM_{2.5}$_OBS) and modeled $PM_{2.5}$ by EXP1, EXP2, and EXP3 experiments, together with the modeled turbulent DC of the three experiments in Beijing (Figure8a) and Xingtai (Figure 8b) from 15 to 23 December. Comparison of the $PM_{2.5}$ modeled by EXP1, EXP2, and EXP3 with observation in Beijing (Figure 8a) shows that the modeled $PM_{2.5}$ by EXP3 was the closest to observation during the whole haze episode, which agreed with the results of regional distribution of EG stage in Figure 4. EXP1 underestimated the $PM_{2.5}$ obviously from 17 to 22 December and this underestimation was even more obvious with the increasing of $PM_{2.5}$. This difference between the modeled and observed $PM_{2.5}$ was the largest during the EG stage of $PM_{2.5}$. AF shortened this difference to a great extent and $PM_{2.5}$ by EXP2 was much closer to the observation than that by EXP1 during EG stage of $PM_{2.5}$. However, it can be seen that there was certain differences between observed and modeled $PM_{2.5}$ by EXP2, illustrating that AF can't completely fill the gap between observed and modeled $PM_{2.5}$. $PM_{2.5}$ by EXP3 shortened this gap further and shows the best agreement with observation, especially during the $PM_{2.5}$ EG stage.

It also can be seen from figure 8a that the DC by EXP1 was about 30-40 m$^2$/s during the EG stage of $PM_{2.5}$, which was about 50% of the 60-70 m$^2$/s on the clear day (15 or 22 December). Obviously, the 50% DC differences between the clear and severe haze days may not be enough to discriminate the difference of turbulent diffusion intensity between extreme stable atmosphere on haze day and unstable atmosphere on clear day, which is the important reason for underestimation of $PM_{2.5}$ EG by EXP1. AF led to notable enhancement of temperature inversion (Figure 7b), significant decrease in turbulent diffusion on $PM_{2.5}$ during EG stage and maximum DC at noon by EXP2 was as low as 14m$^2$/s on 20 December, which decreased about 50% comparing with that by EXP1. Maximum DC at noon by EXP2 on haze day was only about 20% of that on clear day. The maximum DC at noon by EXP3 was lower than 5m$^2$/s on 20 December and at the same time PM$_{2.5}$  af by EXP3 was further increased and it was also much further closer to the

PM$_{2.5}$  observation than  PM$_{2.5}$  af by EXP2.

It can be seen from the comparative study of the temporal changing between DC and PM$_{2.5}$  by

EXP1, EXP2, EXP3 in Beijing that the overestimation of turbulent DC

owning to lack of online calculation of AF and deficient description of the extreme stable stratification by

PBL schemes in atmospheric model led to distinct underestimation of PM$_{2.5}$ EG and peaks when severe haze occurred in Jing-Jin-Ji in China.

The changing trends of DC and PM$_{2.5}$  by the three sensitive experiments in Xingtai (Figure 8b)

shows the similar results with those in Beijing. The PM$_{2.5}$ by EXP3 was also the closest to

 observation, followed by EXP2 and by EXP1 was the worst during the whole haze episode. However during the EG stage of PM$_{2.5}$, the relative contributions on the PM$_{2.5}$

peak values due to AF and DTD showed some difference with those in Beijing. The contributions to PM$_{2.5}$

peaks due to DTD were more important than that by AF in Xingtai. Located at the east foot of

 Taihang Mountains, Xingtai is usually affected by the downhill airflow and temperature inversion in this area is easy to form and strengthened, leading to stronger inversion, weaker turbulent diffusion and more stable atmospheric stratification,  but  this kind of inversion and weak turbulent diffusion derived from local terrain is more difficult to describe by PBL scheme in atmospheric chemical models and likely underestimated. .

Figure 9 shows the diagrammatic sketch of the contributions to the PM$_{2.5}$ of EG stage due to

AF and DTD summarized by the results of Beijing and Xingtai. It can be seen that the DC by EXP1

was 30-35m$^2$/s, DC by EXP2 was 15-17 m$^2$/s, means that AF reduces about 43-57% DC by

EXP1, which led to the rise in simulated PM$_{2.5}$ from 144 ug/m$^3$ by EXP1 to 205 ug/m$^3$

by EXP2 in Beijing, 280 ug/m$^3$ by EXP1 to 360 ug/m$^3$ EXP2 in Xingtai.

This means that AF reduced 20% in Beijing and 25% in Xingtai of simulated PM$_{2.5}$ negative errors.

DC by EXP3 was as low as 4-6 m$^2$/s during EG stage of PM$_{2.5}$, showing the joint effects of

AF and DTD reduced DC to less than 4-6 m$^2$/s, near-zero, we name it as "turbulent intermittent".

The direct results of this "turbulent intermittent" is the further increasing of simulated surface PM$_{2.5}$ based on EXP2. DTD decreases 14% to 20% underestimation of simulated $PM_{2.5}$ and the errors of $PM_{2.5}$  by EXP3 were reduced as low as -11% to 2%.

**4. Conclusions**

Using atmospheric chemical model GRAPES_CUACE, three experiments EXP1, EXP2 and EXP3 were designed to study the reason for the explosive growth of $PM_{2.5}$ mass during a red-alert heavy haze occurred on 15 to 23 December, 2016 in Jing Jin-Ji in China. The contributions to the $PM_{2.5}$ by aerosols feedback and a further decrease in turbulent diffusion coefficient of chemical tracers, representing a compensation for the deficient description of extreme weak turbulent diffusion by PBL scheme in atmospheric models, are studied by analysing the changes of $PM_{2.5}$, surface downward solar radiation flux, wind speed and temperature, diffusion coefficient and the relationships between them .

The study shows that the diffusion coefficient by EXP1 is about 60-70m$^2$/s on clear day and 30-35m$^2$/s on haze day. The 50% difference of the two was not considered enough to discriminate the unstable atmosphere on clear day and extreme stable atmosphere on severe haze day comparing with the differences of direct downward solar radiation between clear and haze days, which is also proved indirectly by the weaker inversion calculated by EXP1 than that of the actual sounding observation. This led to 40-51% underestimation of the $PM_{2.5}$ peaks by EXP1 during the explosive growth stage of $PM_{2.5}$. Online calculation of aerosols radiation feedback reduced 
[revised manuscript text omitted]

Zhong, J.T., Zhang, X.Y., and Wang, Y.Q., Cheng Li., and Dong,Y.S., Heavy aerosol pollution episodes in winter Beijing enhanced by radiative cooling effects of aerosols, Atmos. Res., 2018b, 209, 59-64.

Zhou, C.H., Gong, S., Zhang, X.Y., Liu, H.L., Xue, M., Cao, G.L., An, X.Q., Che, H.Z., Zhang, Y.M., Niu, T., 2012. Towards the improvements of simulating the chemical and optical properties of Chinese aerosols using an online coupled model – CUACE/Aero. Tellus B. 64 (1), 91-102.

Zhou, C., Zhang, X., Gong, S., Wang, Y., Xue, M., 2016. Improving aerosol interaction with clouds and precipitation in a regional chemical weather modeling system. Atmos. Chem. Phys. 16 (1), 145-160.

Table 1 Physics and Chemistry processes in GRAPES_CUACE

| Physics and Chemistry | options | References |
|---|---|---|
| Explicit precipitation | WDM6 | Lim and Hong, 2010 |
| Cumulus clouds | KFETA Scheme | Kain, 2004 |
| Longwave radiation | Goddard | Chou et al., 2001 |
| Shortwave radiation | Goddard | Chou et al., 1998 |
| Surface layer | SFCLAY Schem | Pleim, 2007 |
| Planatory Boundary layer | MRF Schem | Hong et al.,,1996, 2006 |
| Land surface | SLAB Scheme | Kusaka et al., 2001 |
| Gas-phase chemistry | RADM II | Stockwell et al., 1990 |
| Aerosol Scheme | CUACE | Zhou et al., 2012 |
| Aerosol Direct effect | External Mixing | Wang et al., 2015 |
| Aerosol Indirect effect | CAUCE+WDM6 | Zhou et al., 2016 |

Table  2 Sensitive Experiments Design

| Experiments | Description of model Experiments |
|---|---|
| EXP1 | Background experiment: ignoring aerosols radiation and conventional DC of chemical tracers  by PBL scheme in GRAPES_CUACE |
| EXP2 | Sensitive experiment with aerosols radiation feedback online and conventional DC of chemical tracers by PBL scheme in GRAPES_CUACE |
| EXP3 | Sensitive experiment with aerosols radiation feedback online,  only DC of chemical tracers is set as 20% of the conventional DC  calculated by PBL scheme. representing a supposed compensation for the deficient description of extreme weak turbulent diffusion by PBL scheme . DC in physical and dynamic processes was same with EXP1 |

Table 3 VOCs in the emission data

| VOCs | Full name |
|------|-----------|
| ALD | Acetaldehyde and higher aldehydes |
| CH4 | Methane |
| CSL | Cresol and other hydroxy substituted aromatics |
| ETH | Ethane |
| HC3 | Alkanes w/ 2.7x10-13 > kOH < 3.4x10-12 |
| HC5 | Alkanes w/ 3.4x10-12 > kOH < 6.8x10-12 |
| HC7 | w/kOH > 6.8x10-12 |
| HCHO | Formaldehyde |
| ISOP | Isoprene |
| KET | Ketones |
| OL2 | Ethene |
| OLI | Internal olefins |
| OLT | Terminal olefins |
| ORA2 | Acetic and higher acids |
| PAR | Paraffin carbon bond |
| TERPB | Monoterpenes |
| TOL | Toluene and less reactive aromatics |
| XYL | Xylene and more reactive aromatics |


Table 4 Observed and Modeled daily AOD (* stands for shortage of observation )

| Date | Shijiazhuang | | Beijing | | Xianghe | |
|------|------|-------|------|------|------|-------|
| | OBS | MODEL | OBS | MOEL | OBS | MODEL |
| 15 | 0.46 | 0.55 | 0.07 | 0.12 | 0.10 | 0.15 |
| 16 | 0.62 | 0.60 | 0.14 | 0.18 | 0.60 | 0.40 |
| 17 | 1.30 | 1.10 | 0.50 | 0.56 | 1.33 | 1.05 |
| 18 | 1.42 | 1.20 | 0.69 | 0.75 | 0.87 | 0.97 |
| 19 | 1.26 | 1.30 | 0.50 | 0.86 | 0.96 | 0.90 |
| 20 | * | 1.20 | 1.90 | 1.70 | * | 1.50 |
| 21 | * | 0.65 | 1.76 | 1.50 | 1.78 | 1.60 |
| 22 | 0.18 | 0.30 | 0.10 | 0.20 | 0.18 | 0.22 |

Table 5 Observed and Modeled daily SSA (* stands for shortage of observation)

| Date | Shijiazhuang | | Beijing | | Xianghe | |
|------|------|-------|------|------|------|-------|
| | OBS | MODEL | OBS | MOEL | OBS | MODEL |
| 15 | 0.83 | 0.85 | 0.81 | 0.83 | 0.86 | 0.84 |
| 16 | 0.83 | 0.85 | 0.88 | 0.86 | 0.92 | 0.86 |
| 17 | 0.88 | 0.89 | 0.88 | 0.90 | 0.93 | 0.90 |
| 18 | 0.87 | 0.89 | 0.91 | 0.92 | 0.90 | 0.90 |
| 19 | 0.86 | 0.91 | 0.90 | 0.93 | 0.92 | 0.91 |
| 20 | * | 0.90 | * | 0.93 | * | 0.92 |
| 21 | * | 0.88 | 0.93 | 0.93 | * | 0.90 |
| 22 | 0.82 | 0.83 | 0.84 | 0.86 | 0.88 | 0.84 |

**Figure captions**

**Fig.1**  Model domain and location of Jing-Jin-Ji (a),  Features of geographical location and topography of Jing-Jin-Ji (b) (blue dots are the locations of $PM_{2.5}$ observation, red triangles stands for the locations of automatic weather stations, and yellow stars are the two sounding station, black crosses are the CARSNET and AEROSNET stations)

Fig. 2  GPH (shaded, gp10m), Temp (broken black line, K) and Wind (wind bar, m/s) at high (500hPa) and middle (700hPa), and GPH and Wind at low atmosphere (850hPa) and PBL levels (900, 950, 1000hPa) on 00 UTC, 19 December, 2016

Fig. 3 Observed and modeled wind speed and temperature at surface (up) and PBL mean wind speed and temperature (down) by EXP1, EXP2, and EXP3 in Beijing, Xingtai, and average in Jing-Jin-Ji from 15 to 24 December

**Fig.4**  Mean  Observed (OBS_$PM_{2.5}$) and Modeled PM2.5 concentration (μg/m3) of EG stage of $PM_{2.5}$  by EXP1, EXP2, EXP3 ($PM_{2.5}$_EXP1, $PM_{2.5}$_EXP2, and $PM_{2.5}$_ EXP3)

Fig. 5 The mean percentage change of SDSRF (W/m$^2$) due to aerosol (a) and aerosol and DTD (b) of EG stage

**Fig.6**  Profiles of the average temperature changes  in Jing-Jin-Ji due to  AF (K) from 15 to 20 December, 2016.

**Fig.7** Sounding observed and modeled temperature profiles by EXP1 and EXP_td2 during CS (a) and EG stage (b) in Beijing and Xingtai.

**Fig.8** Hourly changing of $PM_{2.5}$_OBS, $PM_{2.5}$_EXP1, $PM_{2.5}$_EXP2, and $PM_{2.5}$_EXP3 (μg/m$^3$), together with the turbulent diffusion coefficient at 950hPa  of the three experiments (DC_EXP1, DC_EXP2, DC_EXP3) from 15 to 22 December, 2016 in Beijing (a) and Xingtai (b)

**Fig.9** The diagrammatic sketch of the contributions to the $PM_{2.5}$ EG due to AF and DTD

[Figure]

**Fig.1** Model domain and location of Jing-Jin-Ji (a), Features of geographical location and topography of Jing-Jin-Ji (b) (blue dots are the locations of PM$_{2.5}$ observation, red triangles stands for the locations of automatic weather stations, and yellow stars are the two sounding station, black crosses are the CARSNET and AEROSNET stations)

**1**Fig. 1 Model domain (a), cities locations and the topography features in Jing-Jin-Ji (b)

[Figure]

Fig. 2 GPH (shaded, gp10m), Temp (broken black line, K) and Wind (wind bar, m/s) at high (500hPa) and middle (700hPa), and GPH and Wind at low atmosphere (850hPa) and PBL levels (900, 950, 1000hPa) on 00 UTC, 19 December, 2016

[Figure]

Fig. 3 Observed and modeled wind speed and temperature at surface (up) and PBL mean wind speed and temperature (down) by EXP1, EXP2, and EXP3 in Beijing, Xingtai, and average in Jing-Jin-Ji from 15 to

24 December

[Figure]

**Fig.4** Mean Observed (OBS_PM$_{2.5}$) and Modeled PM2.5 concentration (μg/m3) of EG stage of PM$_{2.5}$ by EXP1, EXP2, EXP3 (PM$_{2.5}$_EXP1, PM$_{2.5}$_EXP2, and PM$_{2.5}$_ EXP3)

[Figure]

Fig. 5 The mean percentage change of SDSRF (W/m$^2$) due to aerosol (a) and aerosol and DTD (b) of EG

stage

[Figure]

**Fig.6** Profiles of the average temperature changes in Jing-Jin-Ji due to AF (K) from 15 to 20 December,

2016. Fig. 3 Variation of temperature profiles by aerosol radiation (K) from 15 to 20 December, 2016.

[Figure]

**Fig.7** Sounding observed and modeled temperature profiles by EXP1 and EXP2 during CS (a) and EG

stage (b) in Beijing and Xingtai.

[Figure]

首行缩进： 0 字符

**Fig.8** Hourly changing of PM$_{2.5}$_OBS, PM$_{2.5}$_EXP1, PM$_{2.5}$_EXP2, and PM$_{2.5}$_EXP3 (μg/m³), together with the turbulent diffusion coefficient at 950hPa of the three experiments (DC_EXP1, DC_EXP2, DC_EXP3)

from 15 to 22 December, 2016 in Beijing (a) and Xingtai (b)

EXP1, EXP2

[Figure]

**Fig.9** The diagrammatic sketch of the contributions to the PM$_{2.5}$ EG due to AF and DTD

|---|---|---|

字体: (默认) Times New Roman, 10 磅

|---|---|---|

字体: (默认) Times New Roman, 10 磅

|---|---|---|

字体: (默认) Times New Roman, 10 磅

|---|---|---|

字体: (默认) Times New Roman, 10 磅

|---|---|---|

字体: (默认) Times New Roman, 10 磅

|---|---|---|

字体: (默认) Times New Roman, 10 磅

|---|---|---|

字体: (默认) Times New Roman, 10 磅

字体: (默认) Times New Roman, 10 磅

|---|---|---|

字体: (默认) Times New Roman, 10 磅

|---|---|---|

字体: (默认) Times New Roman, 10 磅

|---|---|---|

字体: (默认) Times New Roman, 10 磅

|---|---|---|

字体: (默认) Times New Roman, 10 磅

|---|---|---|

字体: (默认) Times New Roman, 10 磅

|---|---|---|

字体: (默认) Times New Roman, 10 磅

|---|---|---|

字体: (默认) Times New Roman, 10 磅

|---|---|---|

字体: (默认) Times New Roman, 10 磅

|---|---|---|

字体: (默认) Times New Roman, 10 磅

|---|---|---|

字体: (默认) Times New Roman, 10 磅

|---|---|---|

字体: (默认) Times New Roman, 10 磅

|---|---|---|

字体: (默认) Times New Roman, 10 磅

|---|---|---|

字体: (默认) Times New Roman, (中文) +中文正文 (宋体), 10 磅

|---|---|---|

字体: (默认) Times New Roman, (中文) +中文正文 (宋体), 10 磅

|---|---|---|

字体: (默认) Times New Roman, (中文) +中文正文 (宋体), 10 磅

|---|---|---|

字体: (默认) Times New Roman, (中文) +中文正文 (宋体), 10 磅

|---|---|---|

字体: (默认) Times New Roman, 10 磅

|---|---|---|

字体: (默认) Times New Roman, 10 磅

|---|---|---|

字体: (默认) Times New Roman, 10 磅

|---|---|---|

字体: (默认) Times New Roman, 10 磅

---

## Author Response (AR2)

**Comments to the Author:**

Although the authors have addressed the major scientific issues raised by the referees, the paper is technically not yet ready to be accepted for publication. The language of the papers requires major improvements, and should be finally checked out by a native English speaker. My specific comments in this regard are the following:

First, the use of tense, articles and prepositions needs to be checked out and corrected throughout the paper.

**Response:**

Thank you for pointing out the language problems in this paper and giving us the opportunity to correct them. We have asked professional native speaker "LucidPapers" to overall improve the

English of the paper, and after that we proved their corrections in scientific meaning. Please see the edited and clean versions of this paper.

**Second,** there are a number of sentences that are either too long or complicated, or unclear parts, which makes them difficult for the readers. Especially, such sentences can be found on lines 92-93, 107-110, 127-131, 138-141, 164-166 (one cannot say was observed reduction

89%), 180-183 (check out how to write dates), 188-192, 205-209, 234-236, 245-251, 263-268,

285-288, 300-302, 307-308, 320-322, 332-334, 341-343, 348-349, 371-373, 383-384,

398-402, 418-420.

**Response:**

The English of the paper has been polished overall including all the sentences on these lines.

**Third, the paper needs to list proper scientific aim(s). What is written on lines 79-82**

**does not motivate to read this paper in more detail.**

**Response:**

The related context in rewritten in line 92-97 in the edited paper.

Other comments:

There should be a space between the number and unit when given values for quantities

**Response:**

This has been corrected in the revised paper.

Line 121: ... Hong's studies: papers cannot be cited like this

**Response:**

This has been revised as "The turbulent diffusion coefficient (DC) is calculated by the YonSei

University PBL scheme (Hong et al., 2006)" on line 135 in the edited version.

I am not sure the authors use correctly the term "trend" on page 10 and later in the text.

**Response:**

All uses of "trend" (line 151,254,257,264,290,440 in the edited version of this paper) have been examined and corrected.

Line 231: should it be ...overprediction of the temperature … rather than ... temperature positive
errors
**Response:**
This has been corrected as "overestimation of surface temperature (line 270 in the edited version
of this paper)
Line 259: PM2.5 center of... does not sound a right way to express this
**Response:**
This phrase has been revised "The most polluted area with PM2.5 values of 500–700 μg/m3 ···"
on line "305-306" in the edited paper.

[revised manuscript text omitted]

批注 [LP5]: The spatial discretization of what?

批注 [LP6]: Is this what you mean here?

批注 [LP7]: Is this what you mean here?

批注 [LP8]: There is no need to introduce an acronym here, as it is not referred to again in the paper.

mixing of aerosols species (of SS, SD, BC, OC, SF, NI, and AM) and particle size bins is are used in the calculation of aerosols radiation feedbackAF, which wasas introduced and evaluated in detail in previous studies (Wang et al., 2009, 2010, 2015a, 2015b). With this double two-way GRAPES_CUACE model, aerosol-s-radiation–PBL–meteorologyical interactions, as well as aerosol-s-cloud–precipitation interactions, and regional pollution and transportation of $PM_{2.5}$ etc., had have been successfully studied (Wang et al., 2010, 2015a, 2015b; Zhou et al., 2012, 2016; Jiang et al., 2015; Zhang et al., 2018).

The turbulent diffusion coefficient (DC) is calculated by the YonSei University (YSU) PBL scheme (Hong et al., 2006), which is a revised vertical diffusion package based on the nonlocal boundary layer vertical diffusion scheme in a Mmedium-Rrange Fforecast model (MRF) model (Hong et al., 1996). The major ingredient of the revision is the inclusion of an explicit treatment of entrainment processes at the top of the PBL, comparing compared with the MRF PBL scheme. The specific DC calculation method of DC

wasis shown in Hong's studies et al. (????)., This algorithm of DCand has been selected as a standard option for the Medium Rang Forecastin (MRF) Model models (Caplan et al. 1997; Farfán and Zehnder,

2001; Basu, et al., 2002; Bright and Mullen, 2002; Mass et al., 2002) and as well as the Weather Research and Forecasting (WRF) model (Hong et al., 2006) in the National Centers for Environmental Predictions (NCEP) since its establishment.

The model horizontal resolution of the model is adopted as here was 0.15° × 0.15°, to match the resolution of the emission source. Considering the impacts of the interregional transport of pollutants, eEast

China (100°–140°E, 20°–60°N) (fFigure 1a) was set as the model domain, but our discussion mainly focuses mainly on the most polluted area, Jing–Jin–Ji region (the red box frame in fFigure 1a), and for which fFigure 1b showsillustrates the features of geographical location and topography topographical of this regionfeatures. There are two balloon sounding stations, Xingtai and Beijing (yellow stars in fFigure 1b)

in our study area. Xingtai, located in southern Hebei province, at the eastern foot of the Taihang Mountains, and it is influenced by the sinkingdescending airflow from Taihang the Mmountains in winter, and in recent years has frequently been ranked is the most polluted city and the $PM_{2.5}$ concentrations usually ranked the first in China in recently years. The topography of Xingtai and the serious haze pollution it experiences are closely related to its is the typical representative ofsituation on the southern plain of Jing–Jin–Ji. Beijing,

批注 [LP9]: Please cite Hong's work in the correct way, as indicated.

批注 [LP10]: Please check that the changes made here have not affected the intended meaning.

located next to Tianjin and surrounded by Hebei, lies in the transitional zone from the Yan Mountains to its southern plain, and  representing the most polluted areas in the central part of Jing-Jin-Ji.

**2.2 Emissions Inventory**

Based on the Multi-resolution Emissions Inventory for China in 2012 (He et al., 2012), the changes  in East China of  five kinds of emission sources  industrial, domestic, agricultural, natural, and traffic   were obtained from  national  statistical data with respect to industry , energy consumption, road net, and motor vehicles, and updated to 2015 and 2016 . Five reactive gases (SO₂, NO, NO₂, CO, NH₃), 20 volatile organic compounds [VOCsALD, CH₄, CSL, ETH, HC₃, HC₅, HC₈, HCHO, ISOP, KET, NR, OL₂,

OLE, OLI, OLT, ORA₂, PAR, TERPB, TOL, XYL) listed in Table 2], and five aerosols species (BC, OC, SF, NI and fugitive dust) were obtained  via the above emissions data according to the input requirement of the CUACE model. The horizontal grid resolution was 0.15° × 0.15° and there was one emissions data  for each month at hourly intervals.

**2.3 Data**

Hourly observational PM₂.₅ concentration data for more than 1440 surface observational stations (blue dots in Figure 1) from the China National Environmental Monitoring Centre (http://www.cnemc.cn)  during 1523 December 2016 were used to evaluate the model results. The hourly observational  meteorological data, including wind speed and temperature, from 500

surface automatic observation stations  of the China  Meteorological Administration (CMA)

in the Jing-Jin-Ji region (red triangle in Figure 1b), were used  for model validation.

Meteorological balloon sounding data from the CMA at 00 UTC (early morning) and 12 UTC (

dusk,  local time) in Beijing and Xingtai (yellow star in Figure 1b)  during the same period were also used to compare with the modeled results. There  is one AERONET station (Holben et al.,

1998), Xianghe, and two CARSNET stations (Che et al., 2009; 2014; 2015), Beijing and Shijiazhuang, in the Jing-Jin-Ji region (black crosses in Figure 1b). Observed aerosols optical depth (AOD) and single

批注 [LP11]: Please check that the changes made here reflect the intended meaning. Return for further assistance/editing if necessary.

批注 [LP12]: The description here was not very clear and so the editor has made a best guess as to what the intended meaning might be. Please check the suggested changes carefully and return for further assistance/editing if necessary.

scattering albedo (SSA)  data from the three stations  during the same  period were also used  for model evaluation. NCEP 0.25° × 0.25° global analysis  gridded data (https://rda.ucar.edu/datasets/ds083.3) were used as the model's initial and six-hourly lateral boundary  meteorological input fields. The initial values of chemical tracers were obtained according to the five-year mean climatic values. The results of the first 120 hours of the model  were discarded to eliminate the effects of the chemical initial fields.

**2.4  Experimental design**

Both dynamic process of the regional atmosphere and solar radiation have important impacts on  turbulent diffusion and PBL processes. When severe haze occurs, it  has been  showed from the observation study (Zhong et al., 2018) that  surface-level daily direct  radiative exposure is reduced by around 89%  compared with  clean days, suggesting the possibility of a huge difference in  turbulent diffusion between severe haze and clean days. However, it is difficult to distinguish between the two reasons for extreme weak  turbulent diffusion in the  true atmosphere, because of the complicated relationship between atmospheric dynamic and solar radiation. However,  meaningful  results  might be  possible by conducting  sensitivity experiments using an  atmospheric  chemistry model.  Here, three such experiments (EXP1, EXP2, and EXP3 – see Table 3 for descriptions) were designed to discuss the contributing factors to extreme weak turbulence and corresponding $PM_{2.5}$ explosive growth, along with the insufficient description of extremely weak turbulent diffusion by PBL scheme in atmospheric chemistry model.  All other model dynamic process, physical options, and initial input data of the meteorology and chemical tracers were same for the three experiments, i.e., except the differences shown in Table 3. In  EXP3, a further decrease in  turbulent diffusion coefficient (DTD) based on EXP2 was only applied to the DC of chemical tracers in CUACE modethe DC in other physical packages and the dynamic frame of GRAPES_MESO was the same as in EXP1 and EXP2.

批注 [LP13]: Is this what you mean here? Return for further assistance/editing if necessary.

批注 [LP14]: Please check that the changes made here have not affected the intended meaning.

批注 [LP15]: It is not clear to the editor what 'two reasons' are being referred to here. Return for further assistance/editing if necessary.

批注 [LP16]: Please check that the changes made here have not affected the intended meaning.

**3 Results and discussion**

 The studied haze episode began on 15 December 2016. $PM_{2.5}$ began to gather and climb slowly at this time, but  was below 150 ug/m$^3$ in most of Jing-Jin-Ji  from 00:00 UTC  15 to 00:00 UTC  17 December  a period we refer to as the "climbing stage" (CS) of $PM_{2.5}$. From 00:00 UTC  17 to 00:00 UTC 21 December, $PM_{2.5}$ increased rapidly, and  reaching the $PM_{2.5}$ a peak of 400-600 ug/m$^3$ in most of the study area.  We refer to this period  as the "explosive growth  stage" of $PM_{2.5}$.  In this section, we focus mainly  on the  contributions of  AF and  DTD to the $PM_{2.5}$ during this stage.

**3.1 Synoptic background **

The  circulation in the upper atmosphere and the surface-level synoptic system controlling Jing-Jin-Ji  remained relatively stable during the  maintenance of this haze episode. Figure 2 displays the geopotential height , temperature , and wind  in the  upper (500 hPa), middle (700 hPa), and lower  (850 hPa) atmosphere,  as well as PBL levels (900, 950, 1000 hPa),  at 000 UTC 19 December 2016,  to show the  meteorological background . It  can be seen that  the geopotential height in the upper atmosphere (500 hPa) showed zonal circulation in East Asia. There was a horizontal trough north  of Jing-Jin-Ji (black  frame) in the upper and middle atmosphere (500 and 700 hPa), and  the region was controlled by  moderate northwest or west air flow at the bottom of the trough. The temperature and wind fields at 500 and 700 hPa both showed that cold air in the upper and middle atmosphere was weak.  The 850-hPa geopotential height  showed that the subtropical high  in the East Sea was strong  also, Jing-Jin-Ji was in the pressure equalization field to the northwest periphery of the subtropical high and the wind was very weak  at this level due to the block of the subtropical high.  The 900-, 950-, and 100-hPa geopotential heights all showed that Jing-Jin-Ji was located in the pressure equalization field between the "northwest land high"  and southeast subtropical high within the whole PBL, and the land high was weaker than the subtropical high. This resulted in a small pressure gradient, weak and thin wind fields, and a stable  atmospheric situation within the PBL , which  was conducive

批注 [LP17]: Information like this should be given in the figure caption, not the main text.

批注 [LP18]: The editor has assumed this to be a proper noun, i.e., the official name of this sea. If not, please explain what you mean by 'east sea'. Do you in fact mean 'the East China Sea'?

[revised manuscript text omitted]

muchwas minor. On the other hand, the reduction of-in SDSRF due owing to aerosols radiation was already very greatconsiderable, and so the change of-in SDSRF due owing to the increased total-column PM$_{2.5}$ by DTD, would not be so great on a secondary basis. This value of the SDSRF reduction due owing to aerosols and DTD is basically consistent with the 56%--89% difference of observational radiant radiative exposure between clear and haze days at during the same period (Zhong et al., 2018).

**3.4 The Influence of aerosols' reform on the reforming of the local atmosphere atmospheric temperature profile**

Offline and online studies indicated the a reforming of the atmosphere atmospheric temperature profile by owing to aerosols the direct effect of aerosol radiation (Wang et al., 2010, 2015b; Forkel et al., 2012; Gao et al., 2014, 2015; Wang et al., 2014; Gao et al., 2017; Ding et al., 2016). In our previous works (Wang et al., 2015a, 2015b), Ccomposite aerosols mixing of black carbonBC, organic carbonOC, sulfateSF, nitrateNI, dust, ammonium, and sea salt aerosols had beenwas online coupled online coupled into the in GRAPES_CAUCE model. On this basis, in the present study, the changes of-in the mean temperature profile of Jing--Jin--Ji region ofduring daytime due owing to aerosols radiation were calculated from-for 15--to 20 December, 2016 in this work. It can be seen from Figure 6 that aerosols cooled the atmosphere below 750--to 800 hPa, while but warmed the atmosphereit above this height. Considering the PBL height may be as low as several hundreds to one thousand meters when severe hazes occurs in Jing--Jin--Ji (Wang et al., 2015a; Zhong et al., 2017), it may be concluded that the whole PBL and its near upper atmosphere was were cooled by aerosols to a different varying extent during the different stages of this haze process. The aerosols' warming effects of aerosols above 750--850 hPa height were very weak, and the temperature changes differences among different days were also small. However, the aerosols' cooling effects of aerosols shows varied the most between different days differences from the surface to 975 hPa height on different day. The For instance, surface daytime cooling is was about 2.2 K on 19 December, 1.5 K on 18 and 20 December, 1 K on 17 December, and 0.5--0.6 K on 15--to 16 December. This aerosols' cooling effect of aerosols decreased rapidly with the height. The difference of in the cooling rates between the surface and 850 hPa is was 1.8 K on 19 December, 1.3 K on 18 and 20 December, 1 K on 17 December, and 0.3--0.4 K on 15 and 16 December. The difference of in the cooling rates by owing to aerosols between

批注 [LP21]: Please check that the changes made here have not affected the intended meaning. Return for further assistance/editing if necessary.

批注 [LP22]: Please check that the changes made here have not affected the intended meaning. Return for further assistance/editing if necessary.

the surface and the upper PBL  was much bigger during the  explosive growth stage than  the  climbing stage. This may have result in  further intensification of the temperature inversion layer that  already existed during  the haze event, which will be discussed  in the following section.

The vertical sounding meteorological data...

The  meteorological data from the vertical soundings  taken at Beijing and Xingtai  were used to  verify  this change  in the temperature profile  owing to aerosols . Figure 7 shows the vertical temperature profiles of the sounding observation and the modeled temperature profiles  of EXP1 and EXP2 during the climbing stage (Figure 7a) and  explosive growth stage (Figure 7b) at the two stations. The temperature profiles (Figure 7a) show that  the modeled results  of EXP1 and EXP2 both  simulated in part the observed temperature inversion  at Beijing and Xingtai on 15 to 16 December. The  negligible difference between the temperature profiles  of EXP1 and EXP2  indicates that aerosol radiation had very little impact on the temperature profiles and local inversion during the climbing stage. Nevertheless, Figure 7b shows that the observed temperature inversions were obvious stronger and thicker on 18 to 19 December ( explosive growth stage) than those on 15 to 16 (climbing stage), both in Xingtai and Beijing. The temperate profiles  of EXP2 were much closer to the observation results than  those of EXP1; and especially, the temperature inversions were much stronger and also closer to  observation than  those of EXP1. This result  proves that the  correction of local inversions by aerosols during the  PM_{2.5} explosive growth stage was effective.

However, it  can also be seen that the inversions  of EXP2, which included online AF,  were still weaker than  observed  at the two stations. This suggests  there must be other  reasons, besides the online calculation of AF, for the underestimation of the observed extreme strong inversion by the model, which is worthy of  further study.

**3.5  Contributions  of AF and DTD to PM_{2.5} explosive growth**

Turbulent diffusion  is the main  process of gas and particle  exchange from  the surface to  upper atmosphere, and then  removal by  high- altitude transport,

批注 [LP24]: Please check that the suggested change of wording here reflects the intended meaning. Return for further assistance/editing if necessary.

[revised manuscript text omitted]

批注 [LP25]: Please check that the changes made here have not affected the intended meaning. Return for further assistance/editing if necessary.

批注 [LP26]: Please check that the changes made here reflect the intended meaning. Return for further assistance/editing if necessary.

批注 [LP27]: Please check that the changes made here have not affected the intended meaning. Return for further assistance/editing if necessary.

chemistry models  and likely underestimated

Figure 9 is a diagrammatic sketch of the contributions  of AF and DTD summarized by the results at Beijing and Xingtai. It can be seen that the DC of EXP1 was 30–35 m$^2$/s, while that of EXP2 was 15–17 m$^2$/s, meaning  AF reduced the DC by about 43%–57% , which led to the rise in simulated PM$_{2.5}$ from 144 ug/m$^3$ in EXP1 to 205 ug/m$^3$ in EXP2 at Beijing, and from 280 ug/m$^3$ in EXP1 to 360 ug/m$^3$ in EXP2 at Xingtai. This means that AF reduced the underestimation of PM$_{2.5}$ at Beijing and Xingtai by 20%  and 25% , respectively. The DC of EXP3 was as low as 4–6 m$^2$/s during the explosive growth stage , demonstrating the joint effects of AF and DTD reduced the DC to less than 4–6 m$^2$/s, near-zero, which we refer to  as "turbulent intermittence". The direct result of this "turbulent intermittence" was a further increase in the simulated surface PM$_{2.5}$, based on EXP2. DTD reduced the underestimation of simulated PM$_{2.5}$ by 14% to 20%, and the PM$_{2.5}$ errors by EXP3 were reduced to as low as 11% to 2%.

**4. Conclusions**

Using an atmospheric chemistry model, GRAPES_CUACE, three experiments (EXP1, EXP2 and EXP3) were designed to study the reason for the explosive growth of PM$_{2.5}$ mass during a "red-alert" heavy haze event that occurred during 15–23 December 2016 in China's Jing–Jin–Ji region. The contributions of AF and DTD to the PM$_{2.5}$, representing  compensation for the deficient description of extreme weak turbulent diffusion in the PBL scheme of the atmospheric model, were studied by analyzing the changes in PM$_{2.5}$, SDSRF, wind speed and temperature, DC, and the relationships among them, in the three experiments.

Results show that the DC in EXP1 was about 60–70 m$^2$/s on clear day and 30–35 m$^2$/s on haze day. The 50% difference between the two was considered insufficient to separate the unstable atmosphere on clear day and the extreme stable

批注 [LP28]: Please check that the changes made here reflect the intended meaning. Return for further assistance/editing if necessary.

批注 [LP29]: Please check that the changes made here reflect the intended meaning. Return for further assistance/editing if necessary.

atmosphere on severe haze day,  compared with the differences  in direct downward solar radiation between clear and haze days, which  was also  proven indirectly by the weaker inversion of EXP1 than that  from the actual sounding observation. This led to  40%51% underestimation of the $PM_{2.5}$ peaks  in EXP1 during the  $PM_{2.5}$ explosive growth stage. Online calculation of AF reduced  surface and PBL wind speed and cooled the surface and PBL atmosphere. The surface daytime cooling due to aerosol radiation was 1.52.2 K during the explosive growth stage  and 0.50.6 K during the climbing stage . The cooling effect of aerosols decreased rapidly with  height, and this  was a major reason for the strengthening of the temperature inversion during the explosive growth stage . The reduced DC  owing to AF reached 43%57% during the $PM_{2.5}$  explosive growth stage .  The local inversion simulated  in EXP2 was strengthened and closer to the actual sounding observation than that  of EXP1. This resulted in a 20%25% reduction  in the underestimation of $PM_{2.5}$,  with $PM_{2.5}$ errors  in EXP2 being as low as 16 to 11% during the explosive growth stage . The impact on $PM_{2.5}$ owing to AF in the model run was distinct during the explosive growth stage, but minor during the climbing stage, indicating a critical value of 150 ug/m³ of $PM_{2.5}$ leading to an effective AF in online atmospheric chemistry models. However, the local inversion simulated by EXP2 was still weaker than observed, and the $PM_{2.5}$  of EXP2 was still smaller than observed, illustrating  AF could not solve all the $PM_{2.5}$ underestimation problems. In EXP3, the DTD of particles and gas based on EXP2 resulted in a 14%20% lessening of  $PM_{2.5}$ underestimation based on EXP2, and the $PM_{2.5}$ errors of EXP3 were reduced to 11% to 2%.

 The present study illustrates that the PBL scheme in current atmospheric  chemistry models  are probably insufficient for describing the extremely stable atmosphere

批注 [LP30]: This sentence was repeated soon after in the same paragraph. It seemed more relevant there, so this instance has been deleted. Return for further assistance/editing if necessary.

resulting in explosive growth of $PM_{2.5}$ and severe haze in  Jing-–-Jin-–-Ji region.  This may involve  two important reasons:  the absence of  online calculation of AF,  and/or a deficient description of  extreme weak turbulent diffusion by  PBL scheme in the atmospheric  chemistry model. Our study suggests that  online calculation of AF and an improvement in   representation of turbulent diffusion in PBL schemes focus on extreme stable  atmospheric stratification in atmospheric  chemistry 
[revised manuscript text omitted]
 (b) (blue dots are the locations of $PM_{2.5}$ observation, red triangles stands for the locations of automatic weather stations, and yellow stars are the two sounding station, black crosses are the CARSNET and AEROSNET stations)

**Fig. 2.** Geopotential height (color-shaded; gp10m), temperature (dashed black contours; K) and wind (wind bars; m/s) in the (a) upper (500 hPa) and (b) middle (700 hPa) atmosphere, and geopotential height and wind in the (c) lower atmosphere (850 hPa) and (d–f) PBL (900, 950, 1000 hPa), at 0000 UTC 19 December 2016. Fig. 2 GPH (shaded, gp10m), Temp (broken black line, K) and Wind (wind bar, m/s) at high (500hPa) and middle (700hPa), and GPH and Wind at low atmosphere (850hPa) and PBL levels (900, 950, 1000hPa) on 00 UTC, 19 December, 2016

**Fig. 3.** Observed and modeled wind speed and temperature at the surface (upper panels), and the PBL-mean wind speed and temperature (lower panels), from the results of EXP1, EXP2 and EXP3 for Beijing, Xingtai, and the average for Jing–Jin–Ji as a whole, during 15–24 December 2016. Fig. 3 Observed and modeled wind speed and temperature at surface (up) and PBL mean wind speed and temperature (down) by EXP1, EXP2, and EXP3 in Beijing, Xingtai, and average in Jing-Jin-Ji from 15 to 24 December

**Fig. 4.** Mean observed (OBS_PM$_{2.5}$) and modeled $PM_{2.5}$ concentration (µg/m$^3$) of the $PM_{2.5}$ explosive growth stage, from the results of EXP1, EXP2 and EXP3 (PM$_{2.5}$_EXP1, PM$_{2.5}$_EXP2 and PM$_{2.5}$_ EXP3, respectively). Fig.4 Mean Observed (OBS_PM$_{2.5}$) and Modeled PM2.5 concentration (µg/m3) of EG stage of PM$_{2.5}$ by EXP1, EXP2, EXP3 (PM$_{2.5}$_EXP1, PM$_{2.5}$_EXP2, and PM$_{2.5}$_ EXP3)

**Fig. 5.** Mean percentage change in SDSRF (W/m$^2$) owing to (a) aerosols and (b) aerosols+DTD during the explosive growth stage. Fig. 5 The mean percentage change of SDSRF (W/m$^2$) due to aerosol (a) and aerosol and DTD (b) of EG stage

**Fig. 6.** Profiles of average temperature changes in Jing–Jin–Ji owing to AF (K) during 15–20 December

2016.

**Fig. 7.** Sounding-observed and modeled temperature profiles in EXP1 and EXP2 during the (a) climbing stage and (b) explosive growth stage in Beijing and Xingtai.

**Fig. 8.** Hourly change of $PM_{2.5}$_OBS, $PM_{2.5}$_EXP1, $PM_{2.5}$_EXP2, and $PM_{2.5}$_EXP3 ($\mu g/m^3$), together with the DC at 950 hPa of the three experiments (DC_EXP1, DC_EXP2, DC_EXP3) during 15–22 December

2016 in (a) Beijing and (b) Xingtai.

**Fig. 9.** Diagrammatic sketch of the contributions of AF and DTD to the $PM_{2.5}$ explosive growth.

[Figure]

**Fig. 1.** (a) Model domain and location of Jing-Jin-Ji  (b) Geographical location and topography of Jing-Jin-Ji Blue dots are the locations of $PM_{2.5}$ observations; red triangles

批注 [LP33]: In (b), change the labelling to Taihang Mountain**s** and Yanshan Mountain**s** (i.e., plural).

are the locations of automatic weather stations;  yellow stars are the two sounding stations; black crosses are the CARSNET and AEROSNET stations.

[Figure]

Fig. 2.  Geopotential height (color-shaded ; gp10m),  temperature ( dashed black contours ; K) and wind (wind bar ; m/s)  in the (a)  upper (500 hPa) and (b) middle (700 hPa) atmosphere, and  geopotential height and wind  in the (c) lower atmosphere (850 hPa) and (d–f) PBL  (900, 950, 1000 hPa),  at 0000 UTC, 19 December, 2016.

批注 [LP34]: Please label the panels (a) to (f).

[Figure]


**Fig. 3.** Observed and modeled wind speed and temperature at the surface (upper panels), and the PBL -mean wind speed and temperature (lower panels),  from the results of EXP1, EXP2, and EXP3  for Beijing, Xingtai, and the average  for Jing-Jin-Ji as a whole,  during 15 to 24 December 2016.

[Figure]

**Fig. 4.** Mean observed (OBS_PM$_{2.5}$) and modeled PM$_{2.5}$ concentration (μg/m$^3$) of

PM$_{2.5}$ explosive growth stage, from the results of EXP1, EXP2 and EXP3 (PM$_{2.5}$_EXP1, PM$_{2.5}$_EXP2

and PM$_{2.5}$_ EXP3, respectively).

[Figure]

**Fig. 5.** Mean percentage change  in SDSRF (W/m²)  owing to (a) aerosols  and (b) aerosols

+DTD   during the  explosive growth stage.

[Figure]

**Fig. 6.** Profiles of  average temperature change in Jing–Jin–Ji  owing to AF (K)  during 15  20 December 2016.

[Figure]

**Fig. 7.** Sounding —observed and modeled temperature profiles by in EXP1 and EXP2 during the (a)

climbing stageCS (a) and (b) EG explosive growth stage (b) in Beijing and Xingtai.

批注 [LP36]: Insert a space before the bracketed units in the vertical-axis labels.

[Figure]

**Fig. 8.** Hourly  change of PM₂.₅_OBS, PM₂.₅_EXP1, PM₂.₅_EXP2, and PM₂.₅_EXP3 (μg/m³), together with the DC at 950 hPa of the three experiments (DC_EXP1, DC_EXP2, DC_EXP3)  during 15  22 December 2016 in (a) Beijing  and (b) Xingtai .

[Figure]

批注 [LP37]: Replace the tildes (~) with en dashes (–) in the annotation to indicate a range.

**Fig. 9.** Diagrammatic sketch of the contributions  of AF and DTD to the PM$_{2.5}$
explosive growth.